# Data-Faithful Feature Attribution: Mitigating Unobservable Confounders via Instrumental Variables

**Qiheng Sun**[1,2]**, Haocheng Xia**[3]**, Jinfei Liu**[1,2]*

[1]Zhejiang University
[2]Hangzhou High-Tech Zone (Binjiang) Institute of Blockchain and Data Security
[3]Siebel School of Computing and Data Science
University of Illinois Urbana-Champaign
{qiheng_sun,jinfeiliu}@zju.edu.cn, hxia7@illinois.edu

## Abstract

The state-of-the-art feature attribution methods often neglect the influence of unobservable confounders, posing a risk of misinterpretation, especially when it is crucial for the interpretation to remain faithful to the data. To counteract this, we propose a new approach, data-faithful feature attribution, which trains a confounder-free model using instrumental variables. The cluttered effects of unobservable confounders in a model trained as such are decoupled from input features, thereby aligning the output of the model with the contribution of input features to the target feature in the data generation. Furthermore, feature attribution results produced by our method are more robust when focusing on attributions from the perspective of data generation. Our experiments on both synthetic and real-world datasets demonstrate the effectiveness of our approaches.

## 1 Introduction

The increasing complexity and opacity of machine learning (ML) models in real-world applications boost the demand for feature attribution [8]. The feature attribution methods have been developed to help users understand why a model produces certain outputs from specific inputs. For instance, a loan applicant rejected by a bank's decision-making model might seek reasons behind the denial and what changes could potentially reverse the model's decision. Some recent studies [19, 9] have shifted the focus of feature attribution from a traditional *model-centric* perspective to a new perspective that is *data-centric*. Specifically, users may wish to assign importance values to features according to the *data generation process*, referring to the causal relationships through which features influence the target feature. For example, consider a medical application setting where a patient seeks to understand which personal features aggravated the illness and the cooperative impact of all features on the illness. What the patient really wants to know is how all features collaboratively contribute to the illness, rather than one diagnostic model's prediction output. These two aspects of feature attribution align with the concepts of *model fidelity* and *data fidelity*, respectively. Here, model fidelity refers to the attribution being consistent with the output of the explained model, while data fidelity pertains to the attribution being consistent with the data generation process.

SHapley Additive exPlanations (SHAP) [27] and Integrated Gradients (IG) [38] are prevalent representatives of two distinct series of feature attribution methods, each uniquely satisfying critical axioms including sensitivity, implementation invariance, completeness, and symmetry [28]. These properties are essential for ensuring reasonability and fairness in feature attribution. The SHAP-based methods, grounded in game theory and particularly the Shapley value [32], offer interpretations by evaluating all possible combinations of feature contributions in a discrete feature space. In contrast,

---

*Jinfei Liu is the corresponding author.

38th Conference on Neural Information Processing Systems (NeurIPS 2024).

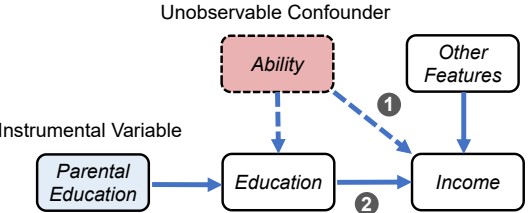

Figure 1: Arrows indicate direct effects. Since *Ability* s not directly observed and is correlated with education, the influence that should be attributed to *Ability* (arrow 1) is erroneously attributed to *Education* (arrow 2) in feature attribution. To fix this issue, *Parental Education* can be used as an instrumental variable for investigating the true impact of *Education* on *Income*.

the IG-based methods, which are an analog of the Aumann-Shapley method [39] from cost-sharing, focus on continuous feature spaces by using path integration over gradients. It is worth noting that even when we aim for interpretations that are faithful to the data, we still rely on a model to predict the target feature values when certain features are selected or excluded in the attribution process. However, when unobservable confounders exist, both SHAP-based and IG-based methods may lead to misunderstandings if applied directly to the widely used predictive model. This is because unobservable confounders, although impacting the output, are entirely overlooked from the process of attribution. Consequently, their influence is instead attributed to other correlated features.

**Motivation example.** As shown in Figure 1, suppose a model predicting personal income includes *Education* as an input feature, and *Ability* serves as a confounder if the model does not incorporate it as an input feature. Because *Ability* has an indirect impact on *Income* through its influence on *Education* and a direct impact on *Income* simultaneously, the existing feature attribution methods tend to incorrectly attach the direct impact of *Ability* on *Income* to the impact of *Education* on *Income* due to their correlation. This concealed correlation may lead to incorrect attribution on the role of educational level in personal income, resulting in an overestimation of the education returns, as demonstrated in our experiments using real datasets (§ 5).

In this paper, we develop a method to eliminate unobservable confounder effects in feature attribution in order to achieve a deeper understanding from the perspective of data fidelity. The instrumental variable method is widely used for causal analysis [24]. It lies in identifying features that directly affect those influenced by confounders, while not having a direct impact on the outcome themselves. By using the instrumental variable to control confounders that influence specific features, any resulting changes in the outcome variable are driven solely by how the instrumental variable alters that feature. For example, in Figure 1, when examining the impact of *Education* on *Income*, a suitable instrumental variable could be the variable *Parental Education* as analyzed in appendix Section F.2. By observing the changes in *Education* resulting from variations in *Parental Education*, we can then discern the true effect of *Education* on *Income*, effectively isolating it from other confounders. Intuitively, the instrumental variable approach can help to mitigate the impact of unobservable confounders in feature attribution. However, the instrumental variable approach mainly focuses on evaluating the isolated impact of individual features influenced by unobservable confounders on the outcome variable, while feature attribution lies in considering the cooperative attribution of features. This means evaluating the combined contribution of *Education* and *Other Features* on *Income*.

To bridge this gap, we propose using a two-stage model training with the instrumental variable that disrupts the association between confounders and other features. Specifically, the model is trained using features re-estimated through instrumental variables and collaborative variables. This ensures that the influence of confounders remains consistent despite variations in feature coalitions. Therefore, the marginal contribution of each feature, which is determined by assessing the impact on the model's output with and without the feature, is not affected by any confounders. Furthermore, the attribution value of each feature, calculated as the average of its marginal contributions across different feature coalitions, remains influenced by confounders. This alignment allows the contribution of input features to the model output to mirror their contribution in data generation to the target feature, thereby facilitating the attribution to be faithful to the data. Feature attribution involves explaining a model output by assigning attribution scores to the input instance. However, our focus is on data-faithful feature attribution, i.e., we are not trying to explain the output of a specific model but trying to explain the target feature through a model.

**Contributions.** To the best of our knowledge, we are the first to identify a crucial issue: unobservable confounders compromise feature attribution, especially when data fidelity is essential. To tackle this challenge, we propose training models free of confounders using instrumental variables, ensuring the feature attribution will remain faithful to the data. We validate the effectiveness of our proposed methods using both real and synthetic datasets, observing that our method achieves up to a 67% relative improvement over the baseline methods in terms of the error of attribution ratio metric in the real dataset.

## 2 Preliminaries

### 2.1 Problem Setup

We aim to quantitatively assess the influence of each input feature on the target feature. This assessment can be viewed as a contribution assignment problem in the context of cooperative game theory [41]. Formally, given an explained input vector of $d$ features $\boldsymbol{x}^* = \{x_1^*, \ldots, x_d^*\}$, a baseline input $\boldsymbol{x}'$, and a model $f : \mathbb{R}^d \to \mathbb{R}$ which approximates the data generation equation for the target feature, our objective is to explain the difference in target feature, i.e., $y^* - y'$, conducting data-faithful attribution for the input features. We assume $\boldsymbol{x}^*$ and $\boldsymbol{x}'$ are of the same dimensionality $d$, and each entry can be either discrete or continuous. We denote by $X$ the set of input features and $Y$ the target feature, partitioning $X$ into two subsets: $\tilde{X}$, which is influenced by unobserved confounders (denoted as $\mathcal{E}$), and $\overline{X}$, the set of other observable features. For clarity and convenience, we use $\boldsymbol{x}, y, \tilde{\boldsymbol{x}}$, $\overline{\boldsymbol{x}}, \epsilon$ to denote possible values within feature sets $X, Y, \tilde{X}, \overline{X}, \mathcal{E}$, respectively. For a given subset of features $\mathcal{S}$, we denote the subset of the original vector of values by using $\mathcal{S}$ as a subscript, e.g., $\boldsymbol{x}_{\mathcal{S}} := \{x_i\}_{i:i \in \mathcal{S}}$.

### 2.2 SHAP-based Attribution

**Shapley Value.** Consider a set of players $\mathcal{N} = \{1, \ldots, d\}$. A *coalition* $\mathcal{S}$ is a subset of $\mathcal{N}$ that cooperates to complete a task. A utility function $\mathcal{U}(\mathcal{S})$ $(\mathcal{S} \subseteq \mathcal{N})$ is the utility of a coalition $\mathcal{S}$ for a task. The *marginal contribution* of player $i$ with respect to a coalition $\mathcal{S}$ is $\mathcal{U}(\mathcal{S} \cup \{i\}) - \mathcal{U}(\mathcal{S})$. Shapley value is the unique metric that satisfies the properties of fair reward allocation, including balance, symmetry, additivity, and zero element [41]. It measures the expectation of marginal contribution by $i$ in all possible coalitions. That is,

$$\mathcal{SV}_i = \frac{1}{|\mathcal{N}|} \sum_{\mathcal{S} \subseteq \mathcal{N} \setminus \{i\}} \frac{\mathcal{U}(\mathcal{S} \cup \{i\}) - \mathcal{U}(\mathcal{S})}{\binom{|\mathcal{N}|-1}{|\mathcal{S}|}}.$$

Computing the exact Shapley value requires enumerating all utilities for all player subsets. Therefore, the computational complexity of exactly calculating the Shapley value is exponential [46].

SHAP [27] utilizes the concept of Shapley values to attribute the contribution of each feature in a model. In the existing SHAP-based methods [27, 19], the definition of utility functions for interpreting an input $\boldsymbol{x}$ can be divided into two categories [5], condition expectation Shapley and intervention Shapley. In condition expectation Shapley, following the assumption that the features are generated according to a distribution $D$, the utility function is defined by $\mathcal{U}^C(\mathcal{S}) = E_D[f(\boldsymbol{x})|\boldsymbol{x}_{\mathcal{S}} = \boldsymbol{x}_{\mathcal{S}}^*]$ [37] based on the condition expectation of the model prediction under feature set $\mathcal{S}$. In intervention Shapley, the utility function is defined by $\mathcal{U}^{\mathcal{I}}(\mathcal{S}) = E_D[f(\boldsymbol{x})|do(\boldsymbol{x}_{\mathcal{S}} = \boldsymbol{x}_{\mathcal{S}}^*)]$ [44] where the operation $do(\boldsymbol{x}_{\mathcal{S}} = \boldsymbol{x}_{\mathcal{S}}^*)$ means we intervene on the features $\mathcal{S}$ in variable $\boldsymbol{x}$ to be the same as the features in $\boldsymbol{x}^*$, while the features outside of $\mathcal{S}$ in $\boldsymbol{x}$ are influenced following the causal relationships of the features [30]. For conciseness, we omit the subscript $D$ in the expectation term in the rest of the paper.

### 2.3 IG-based Attribution

IG is a pivotal method for model attribution [38], particularly well-suited for deep neural networks due to its prerequisite that the model be continuously differentiable. This approach calculates the cumulative gradients along a straight-line path extending from a baseline input $\boldsymbol{x}'$ to the explained input $\boldsymbol{x}^*$. Mathematically, the attribution $\mathcal{IG}_i$ assigned to a particular feature $\boldsymbol{x}_i^*$ for a given input $\boldsymbol{x}^*$

and baseline $x'$ is defined as:

$$\mathcal{IG}_i\left(\boldsymbol{x}^*, \boldsymbol{x}', f\right) = \left(\boldsymbol{x}_i^* - \boldsymbol{x}_i'\right) \int_{\alpha=0}^{1} \frac{\partial f\left(\boldsymbol{x}' + \alpha\left(\boldsymbol{x}^* - \boldsymbol{x}'\right)\right)}{\partial \boldsymbol{x}_i^*} d\alpha. \tag{1}$$

Remarkably, IG shares similarities with the Aumann-Shapley approach [39] and satisfies several essential properties, including linearity, dummy attribution, Affine Scale Invariance (ASI), proportionality, and symmetry [37]. Recent research has enhanced IG through its application to complex models and the refinement of the integrated path [1, 28].

## 3 Misattribution with Unobservable Confounders

In this section, we carry out a theoretical analysis to demonstrate how unobservable confounders mislead the feature attribution of both SHAP-based and IG-based methods. To show the influence of unobservable confounders in feature attribution, we first employ a simple structural equation to characterize the data-generating process, expressed as

$$y = g(\tilde{\boldsymbol{x}}, \overline{\boldsymbol{x}}) + \epsilon,$$

where $g(\tilde{\boldsymbol{x}}, \overline{\boldsymbol{x}})$ can represent any linear or nonlinear continuous relationship involving both $\tilde{\boldsymbol{x}}$ and $\overline{\boldsymbol{x}}$. The equation allows us to clearly recognize the individual contributions of $\tilde{\boldsymbol{x}}$ and $\overline{\boldsymbol{x}}$ to $y$, while also considering the unobserved effects encapsulated in the error term $\epsilon$. Given that $\tilde{X}$ is the set of features influenced by unobservable confounders $\mathcal{E}$, it generally follows that given two data instances $\boldsymbol{x}_1$ and $\boldsymbol{x}_2$, $\mathbb{E}[\epsilon|\tilde{\boldsymbol{x}}_1] \neq \mathbb{E}[\epsilon|\tilde{\boldsymbol{x}}_2]$ when $\tilde{\boldsymbol{x}}_1 \neq \tilde{\boldsymbol{x}}_2$ since $\tilde{X}$ is influenced by $\mathcal{E}$ while $\mathbb{E}[\epsilon|\overline{\boldsymbol{x}}_1] = \mathbb{E}[\epsilon|\overline{\boldsymbol{x}}_2]$ is valid as the feature set $\overline{X}$ is not affected by $\mathcal{E}$.

### 3.1 Example of Errors for Data-Faithful Feature Attribution

We discuss how unobservable confounders introduce errors in data-faithful feature attribution with a toy example of Figure 1 in condition expectation Shapley. We can assume $\epsilon$ as ability, $\tilde{\boldsymbol{x}}$ as the measurement of education level, $\overline{\boldsymbol{x}}$ the work time in a week (i.e., the other variable), and $y$ the weekly income. $\tilde{\boldsymbol{x}}$, $\overline{\boldsymbol{x}}$, and $\epsilon$ each represents a single numerical variable. The data generation equations are defined as follows:

$$\epsilon \sim \text{Uniform}(0,1), \quad \tilde{\boldsymbol{x}} \sim \text{Uniform}(0,1) + \epsilon, \quad \overline{\boldsymbol{x}} \sim \text{Uniform}(0,1), \quad y = \tilde{\boldsymbol{x}} \cdot \overline{\boldsymbol{x}} + \epsilon.$$

In this case, ability influences education, and the three features all have a direct influence on income. Consider a specific data instance $\boldsymbol{x}^* = [\tilde{\boldsymbol{x}}^*, \overline{\boldsymbol{x}}^*] = [1.5, 1]$ we are curious about the contributions of the individual's education level and work hours to their income compared to the features distribution. In this example, it means assigning a value to the individual's education level $\tilde{\boldsymbol{x}}^* = 1.5$ and work time $\overline{\boldsymbol{x}}^* = 1$ to evaluate their contribution to weekly income in comparison to the distribution of education levels and work hours of the population.

First, we conduct feature attribution with a model $f$ that is trained to fit $\mathbb{E}[y|\boldsymbol{x}] = \mathbb{E}[y|\tilde{\boldsymbol{x}}, \overline{\boldsymbol{x}}] = g(\tilde{\boldsymbol{x}}, \overline{\boldsymbol{x}}) + \mathbb{E}[\epsilon|\tilde{\boldsymbol{x}}, \overline{\boldsymbol{x}}]$ which simulates the widely used supervised machine learning model training paradigm in reality. The utility derived from the model is $\mathcal{U}^{\mathcal{C}}(\mathcal{S}) = \mathbb{E}[f(\boldsymbol{x})|\boldsymbol{x}_{\mathcal{S}} = \boldsymbol{x}_{\mathcal{S}}^*] = \mathbb{E}[g(\tilde{\boldsymbol{x}}, \overline{\boldsymbol{x}})|\boldsymbol{x}_{\mathcal{S}} = \boldsymbol{x}_{\mathcal{S}}^*] + \mathbb{E}[\epsilon|\boldsymbol{x}_{\mathcal{S}} = \boldsymbol{x}_{\mathcal{S}}^*]$. The condition expectation Shapley value of $\tilde{\boldsymbol{x}}^*$ is $\mathcal{SV}_{\tilde{\boldsymbol{x}}^*}^{\mathcal{C}} = \frac{1}{2}\{[\mathcal{U}^{\mathcal{C}}(\{\tilde{\boldsymbol{x}}^*\}) - \mathcal{U}^{\mathcal{C}}(\emptyset)] + [\mathcal{U}^{\mathcal{C}}(\{\tilde{\boldsymbol{x}}^*, \overline{\boldsymbol{x}}^*\}) - \mathcal{U}^{\mathcal{C}}(\{\overline{\boldsymbol{x}}^*\})]\}$. Replacing the according utilities, we have $\mathcal{SV}_{\tilde{\boldsymbol{x}}^*}^{\mathcal{C}} = \frac{1}{2}\{\mathbb{E}[g(\tilde{\boldsymbol{x}}^*, \overline{\boldsymbol{x}})] + \mathbb{E}[\epsilon|\tilde{\boldsymbol{x}}^*, \overline{\boldsymbol{x}}] - \mathbb{E}[g(\tilde{\boldsymbol{x}}, \overline{\boldsymbol{x}})] - \mathbb{E}[\epsilon|\tilde{\boldsymbol{x}}, \overline{\boldsymbol{x}}] + \mathbb{E}[g(\tilde{\boldsymbol{x}}^*, \overline{\boldsymbol{x}}^*)] + \mathbb{E}[\epsilon|\tilde{\boldsymbol{x}}^*, \overline{\boldsymbol{x}}^*] - \mathbb{E}[g(\tilde{\boldsymbol{x}}, \overline{\boldsymbol{x}}^*)] - \mathbb{E}[\epsilon|\tilde{\boldsymbol{x}}, \overline{\boldsymbol{x}}^*]\}$.

Then, we conduct feature attribution for $\tilde{\boldsymbol{x}}^*$ with the term which it really contribute to $y$, i.e., $g(\tilde{\boldsymbol{x}}, \overline{\boldsymbol{x}})$. The utility derived is $\overline{\mathcal{U}}^{\mathcal{C}}(\mathcal{S}) = \mathbb{E}[g(\tilde{\boldsymbol{x}}, \overline{\boldsymbol{x}})|\boldsymbol{x}_{\mathcal{S}} = \boldsymbol{x}_{\mathcal{S}}^*]$. According to the definition of condition expectation Shapley, $\overline{\mathcal{SV}}_{\tilde{\boldsymbol{x}}^*}^{\mathcal{C}} = \frac{1}{2}\{[\overline{\mathcal{U}}^{\mathcal{C}}(\{\tilde{\boldsymbol{x}}^*\}) - \overline{\mathcal{U}}^{\mathcal{C}}(\emptyset)] + [\overline{\mathcal{U}}^{\mathcal{C}}(\{\tilde{\boldsymbol{x}}^*, \overline{\boldsymbol{x}}^*\}) - \overline{\mathcal{U}}^{\mathcal{C}}(\{\overline{\boldsymbol{x}}^*\})]\}$. Replacing the accrrording utilities, we have $\overline{\mathcal{SV}}_{\tilde{\boldsymbol{x}}^*}^{\mathcal{C}} = \frac{1}{2}\{\mathbb{E}[g(\tilde{\boldsymbol{x}}^*, \overline{\boldsymbol{x}})] - \mathbb{E}[g(\tilde{\boldsymbol{x}}, \overline{\boldsymbol{x}})] + \mathbb{E}[g(\tilde{\boldsymbol{x}}^*, \overline{\boldsymbol{x}}^*)] - \mathbb{E}[g(\tilde{\boldsymbol{x}}, \overline{\boldsymbol{x}}^*)]\}$.

**Errors in Feature Attribution Values.** The values of each expectation term computed according to the data generation equations are shown in Table 1. Since $\overline{\boldsymbol{x}}$ is independent from $\epsilon$, we have $\mathbb{E}[\epsilon|\tilde{\boldsymbol{x}}, \overline{\boldsymbol{x}}^*] = \mathbb{E}[\epsilon|\tilde{\boldsymbol{x}}, \overline{\boldsymbol{x}}]$ and $\mathbb{E}[\epsilon|\tilde{\boldsymbol{x}}^*, \overline{\boldsymbol{x}}^*] = \mathbb{E}[\epsilon|\tilde{\boldsymbol{x}}^*, \overline{\boldsymbol{x}}]$. By substituting the values for each expected term, we can obtain that $\mathcal{SV}_{\tilde{\boldsymbol{x}}^*}^{\mathcal{C}} = 0.875$, $\overline{\mathcal{SV}}_{\tilde{\boldsymbol{x}}^*}^{\mathcal{C}} = 0.625$, $\mathcal{SV}_{\overline{\boldsymbol{x}}^*}^{\mathcal{C}} = 0.325$, and $\overline{\mathcal{SV}}_{\overline{\boldsymbol{x}}^*}^{\mathcal{C}} = 0.325$. It's

Table 1: Value of each expectation term in $\mathcal{SV}^{\mathcal{C}}_{\tilde{x}^*}$.

| $\mathbb{E}[g(\tilde{x}, \overline{x})]$ | $\mathbb{E}[g(\tilde{x}^*, \overline{x})]$ | $\mathbb{E}[g(\tilde{x}, \overline{x}^*)]$ | $\mathbb{E}[g(\tilde{x}^*, \overline{x}^*)]$ | $\mathbb{E}[\epsilon|\tilde{x}, \overline{x}]$ | $\mathbb{E}[\epsilon|\tilde{x}^*, \overline{x}]$ |
|---|---|---|---|---|---|
| 0.5 | 0.75 | 0.5 | 1.5 | 0.5 | 0.75 |

clear that attribution based on the model trained to fit $\mathbb{E}[y|x]$, which is a common training paradigm in supervised learning, tends to give a wrong attribution value of $\tilde{x}^*$, which is the excessive attribution value of education level in this example. The intuitive reason is that the model associates the direct effect of ability on income with the education level, i.e., the influence of $\epsilon$ is attached to $\tilde{x}$. However, $\mathbb{E}[\epsilon|\tilde{x}^*, \overline{x}] - \mathbb{E}[\epsilon|\tilde{x}, \overline{x}]$ is not actually in the effect of $\tilde{x}^*$ on $y$ because $\tilde{x}$ does not influence $\epsilon$ during the data generation process.

### 3.2 Errors in Feature Attribution with Unobservable Confounders

We analyze the attribution errors when the feature attribution is conducted on a model $f$ trained to fit $\mathbb{E}[y|x] = \mathbb{E}[y|\tilde{x}, \overline{x}]$ in supervised learning for SHAP-based method (Propositions 1 and 2) and IG (Proposition 3), respectively.

**Proposition 1.** *The expected error for marginal contribution of feature $i$ in condition expectation Shapley with model $f$ trained to fit $\mathbb{E}[y|x]$ is $\mathbb{E}[\epsilon|x_{\mathcal{S}\cup\{i\}} = x^*_{\mathcal{S}\cup\{i\}}] - \mathbb{E}[\epsilon|x_{\mathcal{S}} = x^*_{\mathcal{S}}]$, resulting an expected deviation of attribution value by $\Delta\mathcal{SV}_i = \frac{1}{N}\sum_{\mathcal{S}\subseteq\mathcal{N}\setminus\{i\}} \binom{|\mathcal{N}|-1}{|\mathcal{S}|}^{-1}\{\mathbb{E}[\epsilon|x_{\mathcal{S}\cup\{i\}} = x^*_{\mathcal{S}\cup\{i\}}] - \mathbb{E}[\epsilon|x_{\mathcal{S}} = x^*_{\mathcal{S}}]\}$.*

*Proof.* Due to the limited space, please see the appendix for detailed proof. The same to the following propositions. □

**Proposition 2.** *The expected error for marginal contribution of feature $i$ in intervention Shapley with mode $f$ trained to fit $\mathbb{E}[y|x]$ is $\mathbb{E}_D[\epsilon|do(x_{\mathcal{S}\cup\{i\}} = x^*_{\mathcal{S}\cup\{i\}})] - \mathbb{E}[\epsilon|do(x_{\mathcal{S}} = x^*_{\mathcal{S}})]$, resulting an expected deviation of attribution value by $\Delta\mathcal{SV}_i = \frac{1}{N}\sum_{\mathcal{S}\subseteq\mathcal{N}\setminus\{i\}} \binom{|\mathcal{N}|-1}{|\mathcal{S}|}^{-1}\{\mathbb{E}[\epsilon|do(x_{\mathcal{S}\cup\{i\}} = x^*_{\mathcal{S}\cup\{i\}})] - \mathbb{E}[\epsilon|do(x_{\mathcal{S}} = x^*_{\mathcal{S}})]\}$.*

**Proposition 3.** *The expected error for attribution value of feature $i$ using IG with model $f$ trained to fit $\mathbb{E}[y|x]$ is $\Delta\mathcal{IG}_i = (x^*_i - x'_i)\int_{\alpha=0}^{1} \frac{\partial f(x' + \alpha(x^* - x'))}{\partial x_i} - \frac{\partial g(x' + \alpha(x^* - x'))}{\partial x_i} d\alpha$.*

## 4 Mitigating Unobservable Confounders via Instrumental Variables

As demonstrated in Section 3, when it pertains to unobservable confounders, the prevalent feature attribution methods including SHAP and IG inevitably lead to misunderstandings that are not faithful to the data, since they rely on predictive model $f$ trained to fit $\mathbb{E}[y|x]$. This is fundamental because the trained predictive model has already associated the unobservable confounders with the input features. Therefore, it is tempting to ask: *how can we decouple the confounders from their correlations with other features in the used model $f$?*

### 4.1 Motivation of Using Confounder-free Model

One may think a straightforward solution is directly training a model $f$ to fit $g(\tilde{x}, \overline{x})$. Unfortunately, it is nearly impossible since we cannot remove the influence of unobservable confounders in the target feature $y$. To bridge the gap, we provide an alternative solution to train a model that gives the same attribution results for the input features as it is trained to fit $g(\tilde{x}, \overline{x})$. Denote by $\hat{y} = g(\tilde{x}, \overline{x}) + \mathbb{E}[\epsilon]$, and $f$ is trained to fit $\mathbb{E}[\hat{y}|\tilde{x}, \overline{x}]$. The influence of $\tilde{x}$ and $\overline{x}$ on $\hat{y}$ is identical to their impact on $y$, as both are encompassed within $g(\tilde{x}, \overline{x})$.

**Example**. For the toy example in Section 3, the utility calculated with $f$ trained to fit $\mathbb{E}[\hat{y}|\tilde{x}, \overline{x}] = g(\tilde{x}, \overline{x}) + \mathbb{E}[\epsilon]$ is $\hat{\mathcal{U}}^{\mathcal{C}}(\mathcal{S}) = \mathbb{E}[g(\tilde{x}, \overline{x})|x_{\mathcal{S}} = x^*_{\mathcal{S}}] + \mathbb{E}[\epsilon]$. The condition expectation Shapley value of $\tilde{x}^*$ is $\hat{\mathcal{SV}}^{\mathcal{C}}_{\tilde{x}^*} = \frac{1}{2}\{[\hat{\mathcal{U}}(\{\tilde{x}^*\}) - \hat{\mathcal{U}}(\emptyset)] + [\hat{\mathcal{U}}(\{\tilde{x}^*, \overline{x}^*\}) - \hat{\mathcal{U}}(\{\overline{x}^*\})]\}$. By replacing the according utilities, we have $\hat{\mathcal{SV}}_{\tilde{x}^*} = \frac{1}{2}\{\mathbb{E}[g(\tilde{x}^*, \overline{x})] + \mathbb{E}[\epsilon] - \mathbb{E}[g(\tilde{x}, \overline{x})] - \mathbb{E}[\epsilon] + \mathbb{E}[g(\tilde{x}^*, \overline{x}^*)] + \mathbb{E}[\epsilon] - \mathbb{E}[g(\tilde{x}, \overline{x}^*)] - \mathbb{E}[\epsilon]\} = \frac{1}{2}\{\mathbb{E}[g(\tilde{x}^*, \overline{x})] - \mathbb{E}[g(\tilde{x}, \overline{x})] + \mathbb{E}[g(\tilde{x}^*, \overline{x}^*)] - \mathbb{E}[g(\tilde{x}, \overline{x}^*)]\} = \overline{\mathcal{SV}}^{\mathcal{C}}_{\tilde{x}^*}$.

The advantage of attribution based on $\hat{y}$ lies in the term $\mathbb{E}[\epsilon]$ being constant, thereby breaking the association between $\epsilon$ and the input features.

**Proposition 4.** *From the perspective of data generation, the contributions of features in $\tilde{\boldsymbol{x}}$ and $\overline{\boldsymbol{x}}$ to $y$ are equivalent to their contributions to $\hat{y}$. When using the condition expectation Shapley, intervention Shapley, and Integrated Gradients (IG) methods, the attribution values of each feature in $\tilde{\boldsymbol{x}}$ and $\overline{\boldsymbol{x}}$ are identical for both models $f = g(\tilde{\boldsymbol{x}}, \overline{\boldsymbol{x}})$ and $f = g(\tilde{\boldsymbol{x}}, \overline{\boldsymbol{x}}) + \mathbb{E}[\epsilon]$.*

## 4.2 Confounder-free Model Building

With Proposition 4, the problem becomes how to train the model $f$ to fit $\mathbb{E}[\hat{y}|\tilde{\boldsymbol{x}}, \overline{\boldsymbol{x}}]$ now. To achieve this, we introduce the instrumental variables.

**Instrumental Variable.** The features that are used as instrumental variables, denoted as $\Psi$, can be effectively utilized in our model if they satisfy the following three key properties. 1) **relevance**: $\Psi$ should correlate with $\tilde{X}$, ensuring that $\Psi$ can serve as a reliable proxy for these features. 2) **exogeneity**: $\Psi$ should be uncorrelated with the latent confounders $\mathcal{E}$, ensuring that it is not influenced by these unobserved factors. 3) **exclusion restriction**: $\Psi$ should influence the outcome $Y$ solely through its effect on $\tilde{X}$. In other words, apart from its interaction with $\tilde{X}$, $\Psi$ should not have any other direct or indirect pathways affecting $Y$. This ensures that the effect of $\Psi$ on $Y$ can be unambiguously attributed to its relationship with $\tilde{X}$. The effectiveness of IV-SHAP and IV-IG may be reduced when the three assumptions of instrumental variables are violated. However, the extent of this reduction depends on how severely the assumptions are violated.

With the help of instrumental variables, we can establish the following equation by taking the expectation of $y$ given $\overline{\boldsymbol{x}}$ and $\boldsymbol{\psi}$,

$$\mathbb{E}[y|\overline{\boldsymbol{x}}, \boldsymbol{\psi}] = \mathbb{E}[g(\tilde{\boldsymbol{x}}, \overline{\boldsymbol{x}})|\overline{\boldsymbol{x}}, \boldsymbol{\psi}] + \mathbb{E}[\epsilon] = \int g(\tilde{\boldsymbol{x}}, \overline{\boldsymbol{x}}) + \mathbb{E}[\epsilon]dM(\tilde{\boldsymbol{x}}|\overline{\boldsymbol{x}}, \boldsymbol{\psi}),$$

where $\boldsymbol{\psi}$ is a possible value of $\Psi$ and $dM(\tilde{\boldsymbol{x}}|\overline{\boldsymbol{x}}, \boldsymbol{\psi})$ is the conditional distribution of $\tilde{X}$. Given the $T$ training data instances, the optimal parameters of model $f$ trained to fit $g(\tilde{\boldsymbol{x}}, \overline{\boldsymbol{x}}) + \mathbb{E}[\epsilon]$ within the function space $\mathcal{H}$ are identified by minimizing the following objective:

$$\min_{f \in \mathcal{H}} \sum_{t=1}^{T} \mathcal{L}\left(y_t - \int f(\tilde{\boldsymbol{x}}, \overline{\boldsymbol{x}}_t)\, dM(\tilde{\boldsymbol{x}}|\overline{\boldsymbol{x}}_t, \psi_t)\right) \tag{2}$$

where $\mathcal{L}$ represents the loss metric we used to evaluate model performance and $t$ is the index of specific data instance. We provide the training methods for supervised neural network models which are extensively employed in the real world in Section 4.3. Specifically, we discuss their loss functions and gradient computations when the objectives are regression and classification problems. The training steps are inspired by the two-stage training in causal effect estimation [2] and counterfactual prediction [18]. The re-estimated unconfounded values are sampled from the first-stage trained model, the sampling has little influence on the implementation and computation complexity of the second-stage model training. Therefore, the two-stage training process does not limit the method's practical usability. Due to the limited space, we provide details of model training with discrete input features and non-gradient model training in appendix Section D.

## 4.3 Confounder-free Model Training

**Model Training for Continuous Feature Attribution.** In the regression task, where the model $f$ is a neural network trained for forecasting continuous value, it is also denoted as $f_\theta(\tilde{\boldsymbol{x}}, \overline{\boldsymbol{x}})$, where $\theta$ represents the model parameters. As our objective, we adopt a $l_2$ loss function. With the unknown conditional distribution of $\tilde{X}$ given $X$ and $\Psi$, we initially utilize a neural network model, denoted $\hat{M}_\phi$, where $\phi$ is the model parameters, to approximate this distribution. The $l_2$ loss function for determining the optimal model parameters $\theta$ subsequently approximates as per the following equation

$$\mathcal{L}(T; \theta) = |T|^{-1} \sum_{t} \left(y_t - \int f_\theta(\tilde{\boldsymbol{x}}, \overline{\boldsymbol{x}}_t)\, d\hat{M}_\phi(\tilde{\boldsymbol{x}} \mid \overline{\boldsymbol{x}}_t, \boldsymbol{\psi}_t)\right)^2, \tag{3}$$

where the integral term estimates the expected output of $f_\theta$ under the distribution approximated by $\hat{M}_\phi$. By employing the relevant calculations, we ascertain that the gradient of the loss function with respect to the $t^{th}$ training data point is

$$\nabla_\theta \mathcal{L}_t = -2\mathbb{E}_{\hat{M}_\phi(\tilde{\boldsymbol{x}}|\overline{\boldsymbol{x}}_t, \psi_t)} \left[ y_t - f_\theta\left(\tilde{\boldsymbol{x}}, \overline{\boldsymbol{x}}_t\right)\right] \cdot \mathbb{E}_{\hat{M}_\phi(\tilde{\boldsymbol{x}}|\overline{\boldsymbol{x}}_t, \psi_t).} \left[f'_\theta\left(\tilde{\boldsymbol{x}}, \overline{\boldsymbol{x}}_t\right)\right]. \tag{4}$$

In short, our training process comprises two fundamental steps: 1) an instrumental variable method is applied to re-estimate $\tilde{\boldsymbol{x}}$, mitigating the impact of unobservable confounders, and 2) this refined $\tilde{\boldsymbol{x}}$ is utilized to calculate the gradients for $f$.

**Model Training for Discrete Feature Attribution.** In the classification task, where $y$ is a discrete variable representing classes, we adapt the loss function to the multi-class cross-entropy

$$\mathcal{L}(T; \theta) = |T|^{-1} \sum_t \sum_{r=1}^{R} \left( \int y_{t,r} \cdot \ln f_{\theta,r}\left(\tilde{\boldsymbol{x}}, \overline{\boldsymbol{x}}_t\right) d\hat{M}_\phi\left(\tilde{\boldsymbol{x}} \mid \overline{\boldsymbol{x}}_t, \psi_t\right) \right). \tag{5}$$

In this formulation, $y_{t,r}$ represents the true label of the $t^{th}$ data point in the $r^{\text{th}}$ category. $R$ denotes the total number of distinct classes into which the target variable y can be classified. The probability of the model classifying a data point into the $r^{\text{th}}$ category is given by $f_{\theta,r}\left(\tilde{\boldsymbol{x}}, \overline{\boldsymbol{x}}_t\right)$. The gradient calculation for the $t^{th}$ training data point, considering this loss function, is then

$$\nabla_\theta \mathcal{L}_t = \mathbb{E}_{\hat{M}_\phi(\tilde{\boldsymbol{x}}|\overline{\boldsymbol{x}}_t, z_t)} \left[ \sum_{r=1}^{R} \frac{y_{t,r}}{f_{\theta,r}\left(\tilde{\boldsymbol{x}}, \overline{\boldsymbol{x}}_t\right)} \cdot f'_{\theta,r}\left(\tilde{\boldsymbol{x}}, \overline{\boldsymbol{x}}_t\right) \right]. \tag{6}$$

This adaptation of the loss function for discrete target variables ensures that our model can handle classification tasks, effectively optimizing its performance across multiple categories.

**Feature Attribution Computation.** The exact computation of Shapley value and integrated gradients needs huge cost while approximation methods are widely used. Towards practical applications, we further propose a Shapley value approximation method and an integrated gradients approximation method for saving computation costs in Sections E.1 and E.2, respectively. The correlation of input features may affect the data-faithfulness of IV-SHAP and IV-IG. We can combine methods which deal with the correlated input features to the two-stage model to better capture these correlations. For example, on-manifold Shapley [26] can be used to account for feature correlations, while causal Shapley [19] can be applied if the causal structure of the input features is known.

## 5 Experiments

In this section, we present our empirical evaluation in detail. We employ synthetic and real-world datasets to evaluate the faithfulness and robustness against unobservable confounders of feature attributions given by our proposed data-faithful feature attribution methods to prevalent SHAP-based and IG methods. Comparisons of feature attribution for classification problems, feature attribution on non-gradient training models, and the approximation methods of SHAP and IG are given in appendix Sections F.3, F.4, and F.5, respectively. Our code can be found in the repository at `https://github.com/ZJU-DIVER/IV-SHAP`.

### 5.1 Experiments on Synthetic Datasets

We first conducted a data simulation experiment to validate our proposed methods' effectiveness. Synthetic datasets offer an advantage in studying feature attribution, as we can obtain the ground truth of attribution values according to the data generation equation which is unobtainable in most real datasets.

**Dataset Generation Process.** We generated two synthetic datasets, each containing four parts: an unobserved confounder $\epsilon$, a variable $\tilde{\boldsymbol{x}}$ influenced by the unobserved confounder, collaborative variables $\overline{\boldsymbol{x}} = \{\overline{x}_i\}$ $(1 \le i \le 6)$, and the target feature $y$. Notably, dataset A and dataset B share the same $\overline{\boldsymbol{x}}$ and $\psi$. $\epsilon$ is formulated by a uniform variable $v$ and a parameter $\rho$ which controls the noise level. The generation of these features adhered to specific functional relationships, as illustrated in

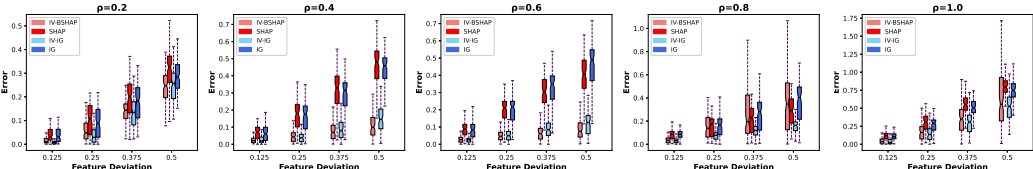

Figure 2: Evaluation results on synthetic Dataset A.

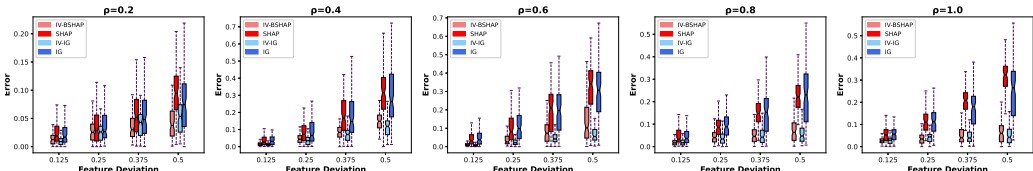

Figure 3: Evaluation results on synthetic Dataset B.

the equations below.

$$v \sim \text{Uniform}(0,1), \quad \overline{x}_i \sim \text{Uniform}(0,1)\ (1 \leq i \leq 6), \quad \psi \sim \text{Uniform}(0,1),$$

Dataset A
$$\begin{cases} \epsilon^A = v \cdot \rho, \\ \tilde{\boldsymbol{x}}^A = \left(\sqrt{\epsilon_a \cdot \psi} + \epsilon^A + \psi^2\right)/3, \\ y^A = \tilde{\boldsymbol{x}}^A + \dfrac{\overline{x}_1^2 + \overline{x}_2 + \sqrt{\overline{x}_3} + \frac{\overline{x}_4^2 + \overline{x}_5 + \sqrt{\overline{x}_6}}{2}}{6} + \epsilon^A, \end{cases}$$

Dataset B
$$\begin{cases} \epsilon^B = \dfrac{\exp(v \cdot \rho - 1)}{\rho}, \\ \tilde{\boldsymbol{x}}^B = \left(\sqrt{\epsilon^B \cdot \psi} + \epsilon^B + \psi^2\right)/3, \\ y^B = \tilde{\boldsymbol{x}}^B \cdot \dfrac{\exp(\overline{x}_1) + \overline{x}_2 + \sqrt{\overline{x}_3} + \frac{\exp(\overline{x}_4) + \overline{x}_5 + \sqrt{\overline{x}_6}}{2}}{6} + \epsilon^B. \end{cases}$$

**Compared Methods.** We utilized two representative feature attribution algorithms, SHAP [27] and IG [38], as the baseline methods. Specifically, we trained a neural network model on synthetic datasets to fit $\mathbb{E}[y|\boldsymbol{x}]$ as the baseline model. Then we applied the two feature attribution methods to attribute contributions for input features. For our proposed approach, we employed the same neural network architecture but trained the model to fit $\mathbb{E}[\hat{y}|\boldsymbol{x}]$ with instrumental variables. Our attribution approaches, applied to the model trained with the instrumental variable, are referred to as IV-SHAP and IV-IG, corresponding to SHAP and IG, respectively. It is worth noting that for the model, the explicitly input data features have no causal relationships among them. Therefore, Causal SHAP [19], Asymmetric SHAP [14], BSHAP [37] and SHAP are equivalent in this context.

**Experimental Results.** We randomly generated 1000 data points based on the data generation equations. We then adjusted features $\tilde{\boldsymbol{x}}$ and $\overline{\boldsymbol{x}}$ of each data point by subtracting a certain value as a baseline input. We conducted experiments with varied subtracted values set at 0.125, 0.25, 0.375, and 0.5. For each data point, we applied IV-SHAP, IV-IG, SHAP, and IG to attribute the contributions of the features. Subsequently, we compared the attribution values of feature $\tilde{\boldsymbol{x}}$ against the ground truth obtained directly from the data generation equations. We observe that the errors in attribution results provided by IV-SHAP and IV-IG are significantly smaller than those of SHAP and IG. The absolute errors in the attribution values of each algorithm for every data point, as compared to the benchmark, are illustrated in Figures 2 and 3.

## 5.2 Experiments on Real-world Datasets

We conducted experiments on two real-world datasets to demonstrate the efficacy of IV-SHAP and IV-IG in practical scenarios where data generation processes are black-box. Initially, we excluded specific features to simulate unobservable confounders. Subsequently, we trained a model using the complete set of features to establish ground-truth attribution values, thereby assessing the robustness and reliability of the compared methods under realistic conditions.

**Real-world Datasets.** The first real dataset we used is the Griliches76 dataset [17, 36], consisting of 758 entries with 20 variables each, gathered from the U.S. labour market. This dataset is extensively used in research to explore the impact of education on income. In the study examining the relationship between the logarithm of weekly earnings (lw) and other features such as educational years (edu), years of work experience (expr), tenure at the current organization (tenure), marital status (mr), residence in the South (rns), and urban residence (smsa), there exists a significant challenge. Ability, as an unobservable confounder, not only directly influences an individual's education level but also their income. Using feature attribution methods like SHAP and IG without accounting for the confounder might incorrectly attribute the effect of ability on income to correlated educational levels. To address this issue, we incorporated the educational years of the mother (medu) as an instrumental variable to affect an individual's education. Additionally, IQ scores (iq) and knowledge in the world of work test (kww) in the dataset, serving as crucial indicators of ability, offer a unique opportunity to approximate the ground truth of the real contribution of each feature.

**Compared Methods.** We assume a decrease in educational years for each individual as baseline inputs and execute a two-phase experiment. Initially, we omit IQ and the world of work test scores, calculating attribution values for IV-SHAP, IV-IG, SHAP, and IG. Note that the process of computing these attribution values using IV-SHAP, IV-IG, SHAP, and IG is consistent with the synthetic dataset experiments. These methods are employed to evaluate the impact of reduced education years. Subsequently, we incorporate IQ and the world of work test scores to train a new model and recalculate attribution values using SHAP and IG. Due to the absence of a real-world benchmark in the reality dataset, we adopt the attribution results from the model, which includes unobservable confounders in its training process, as our benchmark for comparison.

**Evaluation Metric.** Denote the average attribution ratio of IV-SHAP by $\text{EAR}_{\text{IVSHAP}} = \frac{1}{n}\sum_{i=1}^{n}|\frac{\text{IVSHAP}_i}{lw_i}|$ where $\text{IVSHAP}_i$ refers to the educational attribution value for the $i^{th}$ data point, calculated by the IV-SHAP method. $\text{IVSHAP}_i$ represents the extent to which changes in educational years influence the income in the $i^{th}$ data point. $lw_i$ is the income for the $i^{th}$ data point. $\text{EAR}_{\text{IVSHAP}}$ is the average of the absolute values of the ratios between the educational attribution values and income across all data points. This measure provides a comprehensive quantification of the impact of educational years on income. The average attribution ratio for reduced educational years, calculated by SHAP in the model trained with IQ and the world of work test scores, is denoted as $\text{EAR}_{\text{BMSHAP}} = \frac{1}{n}\sum_{i=1}^{n}|\frac{\text{BMSHAP}_i}{lw_i}|$ where $\text{BMSHAP}_i$ represents the benchmark attribution of education on income, incorporating IQ and the world of work test. We then compute the absolute relative error between the attributions of IV-SHAP and the benchmark using the formula $|\frac{\text{EAR}_{\text{IVSHAP}}-\text{EAR}_{\text{BMSHAP}}}{\text{EAR}_{\text{BMSHAP}}}|$. For the SHAP algorithm, $\text{EAR}_{\text{BMSHAP}}$ still serve as a benchmark for $\text{EAR}_{\text{SHAP}}$. For the attribution values calculated by IV-IG and IG, we use the average attribution ratio obtained by the IG algorithm on the model that includes IQ and kww as inputs as the benchmark.

**Experimental Results.** The experimental results are shown in Table 2, which demonstrates that our methods can significantly reduce the attribution error. The values in the table represent the mean of five independent runs, with the standard deviation following each mean.

Table 2: Relative error of each attribution algorithm.

| YEAR | 1 | 2 | 3 | 4 | 5 |
|---|---|---|---|---|---|
| SHAP | $0.566 \pm 0.041$ | $0.569 \pm 0.053$ | $0.569 \pm 0.040$ | $0.548 \pm 0.047$ | $0.552 \pm 0.038$ |
| IV-SHAP | $0.184 \pm 0.032$ | $0.162 \pm 0.026$ | $0.172 \pm 0.025$ | $0.157 \pm 0.019$ | $0.146 \pm 0.021$ |
| IG | $0.554 \pm 0.044$ | $0.582 \pm 0.052$ | $0.583 \pm 0.044$ | $0.467 \pm 0.047$ | $0.538 \pm 0.043$ |
| IV-IG | $0.178 \pm 0.025$ | $0.152 \pm 0.020$ | $0.165 \pm 0.028$ | $0.149 \pm 0.023$ | $0.135 \pm 0.017$ |

**Empirical Analysis.** The second real dataset we use is the Angrist and Krueger dataset [3], which is an American census dataset consisting of statistical data on people born in specific years, including variables such as age (AGE), an education level (EDUC), weekly wage (LWKLYWGE), marital status (MARRIED), and race (RACE). We employed this dataset to examine the confounder effects of the ability on the attribution of education for income. In this dataset, we used the quarter of birth as an instrumental variable for years of education. The rationale behind this is the compulsory education laws in various states, which typically mandate schooling until the age of 16. Students born early in the year often start school later, leading to systematic differences in educational duration based on birth quarter. However, this dataset lacks measures of intelligence or work capability to assess the factor of ability, so we cannot conduct the experiment like the previous one. Nevertheless, the average attribution ratio of one additional year, calculated by SHAP, IV-SHAP, IG, and IV-IG, are 0.0218, 0.0206, 0.0218, and 0.0207, respectively. This aligns with the observation that people may overestimate educational returns because of neglecting the confounder ability in [6].

# 6 Limitations

Despite the strengths of our approach, there are several limitations to consider which are shown as follows:

- **Dependence on the Availability of Instrumental Variables**: Our approach assumes the presence of suitable instrumental variables for features affected by unobserved confounders. However, in practical scenarios, finding appropriate instrumental variables can be challenging sometimes. For further information on identifying instrumental variables, refer to works such as [4], [10], and [23].

- **Linearity Assumption in Theoretical Derivations**: Our theoretical derivations are based on the assumption that the influence of unobserved confounders on the target features is linear. This assumption does not hold in all real-world situations. Nevertheless, in our experiments with real datasets in Section 5.2, our attribution method showed significant improvements over existing methods, even when the influence of unobservable confounders on features was non-linear.

These limitations highlight areas for future research, particularly in developing methods that do not rely on the availability of instrumental variables and that can give theoretical analysis of non-linear effects of unobserved confounders. Addressing these aspects can enhance the practicality and applicability of our methods.

# 7 Conclusion

In this paper, we focus on addressing the effects of unobservable confounders in feature attribution, emphasizing feature attribution being faithful to data. The proposed method improves the understanding of the causal factors driving an outcome variable, going beyond standard attribution scores that simply describe predictive models. Our approach of training confounder-free models using instrumental variables effectively isolates the impact of confounders, enhancing the robustness of data-faithful feature attribution results. Our validations using real and synthetic datasets confirm the effectiveness of the proposed methods. For future work, we intend to develop methods that do not rely on the availability of instrumental variables and that can provide a theoretical analysis of the non-linear effects of unobserved confounders. For the broader impacts of the paper, please see Section A in the appendix due to the limited space.

## Acknowledgment

The authors would like to thank the anonymous reviewers for their helpful comments. This work was supported in part by the National Key RD Program of China (2021YFB3101100), NSFC grants (62102352, U23A20306), The Zhejiang Province Pioneer Plan (2024C01074), and NSF grant (CNS-2125530).

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

# Appendix

In the appendix of our paper, we provide comprehensive additional content. We discuss the lbroader impacts of the paper in Section A, respectively. Section B reviews the related works. In Section D, we delve into the computation of gradients when dealing with discrete features and discuss training methods for non-neural network models. Following this, Section E presents the algorithm for computing Shapley values optimized using confidence intervals, along with an error analysis for unbiased sampling in integrated gradients. Subsequently, Section F offers supplementary material related to our experimental procedures. This includes a detailed analysis of the characteristics of our experimental dataset, justifying our experimental design. Additional results from classification experiments and non-neural network models are also provided.

## A    Broader Impacts

While we believe our paper has many positive social impacts, we think it can particularly affect:

- **Fairness and Equity in Automated Systems**: Reduces biases caused by unobservable confounders in feature attribution, promoting fairness in systems like credit scoring and hiring. This helps to build trust in these systems and supports fair decision-making in various areas.
- **Improved Decision-Making in Healthcare**: Our method enables more accurate identification of factors affecting patient outcomes, leading to better diagnosis, treatment plans, and personalized medicine. Healthcare professionals can make better decisions, which improves patient care and outcomes.

We do not think our paper has any negative social impacts.

## B    Related Work

In this section, we first introduce seminal works that have a significant influence on the feature attribution domain. This is followed by an exploration of works integrating causal knowledge into feature attribution. Additionally, we introduce the widespread presence of confounders in machine learning. Finally, we discuss advancements in computational optimization for SHAP-based methods. For a more detailed survey of feature attribution, please see [12].

### B.1    Classic Feature Attribution Techniques

LIME (Local Interpretable Model-agnostic Explanations) facilitates the understanding of individual predictions of complex models by creating explanatory models [31]. It reveals the impact of features on predictions by perturbing the input and observing the resultant changes in output. DeepLIFT (Deep Learning Important FeaTures) offers a method for assessing feature importance in deep neural networks by comparing the activation of each feature against a reference activation, proving particularly effective in interpreting deep learning models [33]. SmoothGrad enhances visual interpretations of gradient-based methods by applying multiple small random perturbations to the input data and averaging the gradients of these perturbations [35]. Meanwhile, researchers have increasingly recognized that for interpretability methods to be effective and credible, they need to satisfy axiomatic properties. SHAP (SHapley Additive exPlanations) employs Shapley values from cooperative game theory to measure feature contributions, offering a model-agnostic approach with broad applicability [27]. It adheres to desirable allocation properties, ensuring both consistency and equity in attributing feature influence on predictions. Meanwhile, Integrated Gradients (IG) calculates feature importance through the integration of gradients along a straight path from a baseline to the input, making it ideal for scenarios with continuous features and differentiable models [38].

### B.2    Causal Feature Attribution Techniques

In the evolving field of feature attribution, the significance of causal relationships for data-faithful interpretations is increasingly recognized [21]. Asymmetric Shapley values (ASVs) are developed

to infuse causal understanding into model explanations [14]. They achieve this by modifying the symmetry axiom in the Shapley value framework, allowing for the inclusion of causal relationships. Notably, ASVs can provide insights even without a complete causal graph. Causal Shapley values stand out in their capacity to distinguish between direct and indirect feature impacts on model predictions, offering a profound understanding of data generation [19]. Shapley Flow distinguishes itself by evaluating the entire causal graph, attributing influence across its edges rather than focusing solely on nodes [40]. Recursive Shapley Value (RSV) presents a specialized approach for graphical models, quantifying the propagation of changes from source nodes throughout the graph [34]. However, despite the advancements made by these methods in considering the causal relationships between model input features, these methods overlook the impact of unobservable confounders on feature attribution.

### B.3 Confounders in Machine Learning

Researchers have recently begun exploring methods to identify and adjust for confounders in algorithmic models to enhance decision-making quality [22, 42]. Gao et al. [15] identify that pre-trained graph neural networks perform better on pruned graphs than on full graphs due to confounders and introduce Robust Causal Graph Representation Learning (RCGRL) to effectively address this issue by eliminating confounders. Zhang et al. [45] introduce a method for causal imitation learning in the presence of unobservable confounders, featuring a graphical criterion to evaluate its feasibility despite partially observed decision variables behind expert actions. Deep Sequential Weighting (DSW) is proposed for estimating individual treatment effects in healthcare, accounting for time-varying hidden confounders using deep learning [25]. In confounded sequential decision-making, Xu et al. [43] study introduces an instrumental variable (IV) method for off-policy evaluation (OPE) to estimate policy returns accurately in infinite horizon settings.

### B.4 Approximation of SHAP-based Methods

TreeSHAP [26], for tree-based models, enhances SHAP value calculation efficiency by utilizing tree structures to skip redundant feature combination evaluations. Dynamic Shapley, as discussed in the paper by [47], focuses on dealing with scenarios where the players may change. Kernel SHAP [27], suitable for various models, approximates SHAP values by sampling in the feature space and assessing the impact of different feature combinations. Among the recent advancements in optimizing SHAP computation are TMC (Truncated Monte Carlo) [16] and FastSHAP [20], each offering unique approaches to enhance efficiency. TMC employs a truncation technique for rapid, biased sampling approximations. FastSHAP is biased too, employing a pre-trained auxiliary model, speeds up SHAP value prediction. Moreover, unlike methods that approximate Shapley values through sampling, its ability to accurately estimate Shapley values does not improve with more samples, as the precision of the auxiliary model is predetermined upon training. Recently, researchers have proposed one Shapley value approximation method based on the complementary contribution which can be adapted to the general class of feature attribution scenarios [46].

## C Proofs

### C.1 Proof of Proposition 1

*Proof.* From the perspective of the data generation process for $y$, the marginal contribution of feature $i$ is exclusively linked to the function $g$. Thus, the marginal contribution of feature $i$ in condition expectation Shapley for the target feature generation is $\overline{\mathcal{U}}^{\mathcal{C}}(\mathcal{S} \cup \{i\}) - \overline{\mathcal{U}}^{\mathcal{C}}(\mathcal{S}) = \mathbb{E}[g(\tilde{\boldsymbol{x}}, \overline{\boldsymbol{x}})|\boldsymbol{x}_{\mathcal{S} \cup \{i\}} = \boldsymbol{x}_{\mathcal{S} \cup \{i\}}] - \mathbb{E}[g(\tilde{\boldsymbol{x}}, \overline{\boldsymbol{x}})|\boldsymbol{x}_{\mathcal{S}} = \boldsymbol{x}_{\mathcal{S}}]$. For the model trained to fit $\mathbb{E}[y|\tilde{\boldsymbol{x}}, \overline{\boldsymbol{x}}]$, we have $\mathbb{E}[y|\tilde{\boldsymbol{x}}, \overline{\boldsymbol{x}}] = \mathbb{E}[g(\tilde{\boldsymbol{x}}, \overline{\boldsymbol{x}})|\tilde{\boldsymbol{x}}, \overline{\boldsymbol{x}}] + \mathbb{E}[\epsilon|\tilde{\boldsymbol{x}}, \overline{\boldsymbol{x}}] = g(\tilde{\boldsymbol{x}}, \overline{\boldsymbol{x}}) + \mathbb{E}[\epsilon|\tilde{\boldsymbol{x}}]$. Therefore, given explained input $\boldsymbol{x}$, the marginal contribution of a particular feature $i$ with $\mathcal{S}$ in condition expectation Shapley derived with model $f$ can be represented as follows $\mathcal{U}^{\mathcal{C}}(\mathcal{S} \cup \{i\}) - \mathcal{U}^{\mathcal{C}}(\mathcal{S}) = \mathbb{E}[f(\boldsymbol{x})|\boldsymbol{x}_{\mathcal{S} \cup \{i\}} = \boldsymbol{x}_{\mathcal{S} \cup \{i\}}] - \mathbb{E}[f(\boldsymbol{x})|\boldsymbol{x}_{\mathcal{S}} = \boldsymbol{x}_{\mathcal{S}}] = \mathbb{E}[g(\tilde{\boldsymbol{x}}, \overline{\boldsymbol{x}})|\boldsymbol{x}_{\mathcal{S} \cup \{i\}} = \boldsymbol{x}_{\mathcal{S} \cup \{i\}}] + \mathbb{E}[e|\boldsymbol{x}_{\mathcal{S} \cup \{i\}} = \boldsymbol{x}_{\mathcal{S} \cup \{i\}}] - \mathbb{E}[g(\tilde{\boldsymbol{x}}, \overline{\boldsymbol{x}})|\boldsymbol{x}_{\mathcal{S}} = \boldsymbol{x}_{\mathcal{S}}] - \mathbb{E}[e|\boldsymbol{x}_{\mathcal{S}} = \boldsymbol{x}_{\mathcal{S}}]$. Thus, the marginal contribution calculated by the model $f$ which is trained to fit $\mathbb{E}[y|\tilde{\boldsymbol{x}}, \overline{\boldsymbol{x}}]$ includes an error term $\mathbb{E}_D[\epsilon|\boldsymbol{x}_{\mathcal{S} \cup \{i\}} = \boldsymbol{x}_{\mathcal{S} \cup \{i\}}] - \mathbb{E}[\epsilon|\boldsymbol{x}_{\mathcal{S} \cup \{i\}} = \boldsymbol{x}_{\mathcal{S} \cup \{i\}}]$, arises from the model's reliance on the correlations within features $\tilde{X}$ and $\mathcal{E}$. By averaging the errors in the marginal contributions of

feature $i$ with all possible cooperate coalitions, we can get the expected deviation of attribution value by $\Delta \mathcal{SV}_i = \frac{1}{\mathcal{N}} \sum_{\mathcal{S} \subseteq \mathcal{N} \setminus \{i\}} \binom{|\mathcal{N}|-1}{|\mathcal{S}|}^{-1} \{\mathbb{E}_D[\epsilon | \boldsymbol{x}_{\mathcal{S} \cup \{i\}} = \boldsymbol{x}^*_{\mathcal{S} \cup \{i\}}] - \mathbb{E}[\epsilon | \boldsymbol{x}_{\mathcal{S}} = \boldsymbol{x}^*_{\mathcal{S}}]\}$. $\qquad \square$

## C.2 Proof of Proposition 2

*Proof.* From the data generation perspective, the marginal contribution of feature $i$ in intervention Shapley for the target feature generation should be $\overline{\mathcal{U}}^{\mathcal{I}}(\mathcal{S} \cup \{i\}) - \overline{\mathcal{U}}^{\mathcal{I}}(\mathcal{S}) = \mathbb{E}[g(\tilde{\boldsymbol{x}}, \overline{\boldsymbol{x}})|do(\boldsymbol{x}_{\mathcal{S} \cup \{i\}} = \boldsymbol{x}_{\mathcal{S} \cup \{i\}})] - \mathbb{E}[g(\tilde{\boldsymbol{x}}, \overline{\boldsymbol{x}})|do(\boldsymbol{x}_{\mathcal{S}} = \boldsymbol{x}_{\mathcal{S}})]$. However, for the intervention Shapley, the marginal contribution derived with a model trained to fit $\mathbb{E}[y|\tilde{\boldsymbol{x}}, \overline{\boldsymbol{x}}]$ is $\mathcal{U}^{\mathcal{I}}(\mathcal{S} \cup \{i\}) - \mathcal{U}^{\mathcal{I}}(\mathcal{S}) = \mathbb{E}[f(\boldsymbol{x})|do(\boldsymbol{x}_{\mathcal{S} \cup \{i\}} = \boldsymbol{x}_{\mathcal{S} \cup \{i\}})] - \mathbb{E}[f(\boldsymbol{x})|do(\boldsymbol{x}_{\mathcal{S}} = \boldsymbol{x}_{\mathcal{S}})] = \mathbb{E}[g(\tilde{\boldsymbol{x}}, \overline{\boldsymbol{x}})|do(\boldsymbol{x}_{\mathcal{S} \cup \{i\}} = \boldsymbol{x}_{\mathcal{S} \cup \{i\}})] + \mathbb{E}[\epsilon|do(\boldsymbol{x}_{\mathcal{S} \cup \{i\}} = \boldsymbol{x}_{\mathcal{S} \cup \{i\}})] - \mathbb{E}[g(\tilde{\boldsymbol{x}}, \overline{\boldsymbol{x}})|do(\boldsymbol{x}_{\mathcal{S}} = \boldsymbol{x}_{\mathcal{S}})] - \mathbb{E}_D[\epsilon|do(\boldsymbol{x}_{\mathcal{S}} = \boldsymbol{x}_{\mathcal{S}})]$. Thus, we have the expected error for marginal contribution of feature $i$ in intervention Shapley with mode $f$ trained to fit $\mathbb{E}[y|\boldsymbol{x}]$ is $\mathbb{E}_D[\epsilon|do(\boldsymbol{x}_{\mathcal{S} \cup \{i\}} = \boldsymbol{x}^*_{\mathcal{S} \cup \{i\}})] - \mathbb{E}_D[\epsilon|do(\boldsymbol{x}_{\mathcal{S}} = \boldsymbol{x}^*_{\mathcal{S}})]$. By averaging the errors in all the marginal contributions of feature $i$ with all possible cooperative coalitions, we can get the expected deviation of attribution value by $\Delta \mathcal{SV}_i = \frac{1}{\mathcal{N}} \sum_{\mathcal{S} \subseteq \mathcal{N} \setminus \{i\}} \binom{|\mathcal{N}|-1}{|\mathcal{S}|}^{-1} \{\mathbb{E}_D[\epsilon|do(\boldsymbol{x}_{\mathcal{S} \cup \{i\}} = \boldsymbol{x}^*_{\mathcal{S} \cup \{i\}})] - \mathbb{E}_D[\epsilon|do(\boldsymbol{x}_{\mathcal{S}} = \boldsymbol{x}^*_{\mathcal{S}})]\}$. $\qquad \square$

## C.3 Proof of Proposition 3

*Proof.* For IG, where $\epsilon$ is the unobservable confounder correlated with $\boldsymbol{x}$, the derivative $\frac{\partial f}{\partial \boldsymbol{x}}$ is likely to incorporate the effect of $\epsilon$ on $\boldsymbol{x}$, as it is trained to fit $g(\boldsymbol{x}) + \epsilon$. Consequently, the following inequality typically holds $\frac{\partial f}{\partial \boldsymbol{x}} \neq \frac{\partial g}{\partial \boldsymbol{x}}$. Therefore, the attribution $\mathcal{IG}_i(\boldsymbol{x}, \boldsymbol{x}', f)$ derived from the predictive model $f$ generally differs from the attribution $\mathcal{IG}_i(\boldsymbol{x}, \boldsymbol{x}', g)$ that should be obtained based on the actual data generation process. When we accumulate the difference of $\frac{\partial f}{\partial \boldsymbol{x}}$ and $\frac{\partial g}{\partial \boldsymbol{x}}$ in the path, we can get the error for attribution value of feature $i$ using IG with model $f$ trained to fit $\mathbb{E}[y|\boldsymbol{x}]$ is $\Delta \mathcal{IG}_i = (\boldsymbol{x}^*_i - \boldsymbol{x}'_i) \int_{\alpha=0}^1 \frac{\partial f(\boldsymbol{x}' + \alpha(\boldsymbol{x}^* - \boldsymbol{x}'))}{\partial \boldsymbol{x}_i} - \frac{\partial g(\boldsymbol{x}' + \alpha(\boldsymbol{x}^* - \boldsymbol{x}'))}{\partial \boldsymbol{x}_i} d\alpha$.

$\qquad \square$

## C.4 Proof of Proposition 4

*Proof.* The marginal contribution of a particular feature $i$ with $\mathcal{S}$ in condition expectation Shapley derived with model $f = g(\tilde{\boldsymbol{x}}, \overline{\boldsymbol{x}}) + \mathbb{E}[\epsilon]$ can be represented as follows $\mathcal{U}^{\mathcal{C}}(\mathcal{S} \cup \{i\}) - \mathcal{U}^{\mathcal{C}}(\mathcal{S}) = \mathbb{E}[f(\boldsymbol{x})|\boldsymbol{x}_{\mathcal{S} \cup \{i\}} = \boldsymbol{x}_{\mathcal{S} \cup \{i\}}] - \mathbb{E}[f(\boldsymbol{x})|\boldsymbol{x}_{\mathcal{S}} = \boldsymbol{x}_{\mathcal{S}}] = \mathbb{E}[g(\tilde{\boldsymbol{x}}, \overline{\boldsymbol{x}})|\boldsymbol{x}_{\mathcal{S} \cup \{i\}} = \boldsymbol{x}_{\mathcal{S} \cup \{i\}}] + \mathbb{E}[e] - \mathbb{E}[g(\tilde{\boldsymbol{x}}, \overline{\boldsymbol{x}})|\boldsymbol{x}_{\mathcal{S}} = \boldsymbol{x}_{\mathcal{S}}] - \mathbb{E}[e] = \mathbb{E}[g(\tilde{\boldsymbol{x}}, \overline{\boldsymbol{x}})|\boldsymbol{x}_{\mathcal{S} \cup \{i\}} = \boldsymbol{x}_{\mathcal{S} \cup \{i\}}] - \mathbb{E}[g(\tilde{\boldsymbol{x}}, \overline{\boldsymbol{x}})|\boldsymbol{x}_{\mathcal{S}} = \boldsymbol{x}_{\mathcal{S}}]$. Thus, the attribution are identical for models $f = g(\tilde{\boldsymbol{x}}, \overline{\boldsymbol{x}})$ and $f = g(\tilde{\boldsymbol{x}}, \overline{\boldsymbol{x}}) + \mathbb{E}[\epsilon]$ in condition expectation Shapley.

The marginal contribution derived with a model trained to fit $f = g(\tilde{\boldsymbol{x}}, \overline{\boldsymbol{x}}) + \mathbb{E}[\epsilon]$ is $\mathcal{U}^{\mathcal{I}}(\mathcal{S} \cup \{i\}) - \mathcal{U}^{\mathcal{I}}(\mathcal{S}) = \mathbb{E}[f(\boldsymbol{x})|do(\boldsymbol{x}_{\mathcal{S} \cup \{i\}} = \boldsymbol{x}_{\mathcal{S} \cup \{i\}})] - \mathbb{E}[f(\boldsymbol{x})|do(\boldsymbol{x}_{\mathcal{S}} = \boldsymbol{x}_{\mathcal{S}})] = \mathbb{E}[g(\tilde{\boldsymbol{x}}, \overline{\boldsymbol{x}})|do(\boldsymbol{x}_{\mathcal{S} \cup \{i\}} = \boldsymbol{x}_{\mathcal{S} \cup \{i\}})] + \mathbb{E}[\epsilon] - \mathbb{E}[g(\tilde{\boldsymbol{x}}, \overline{\boldsymbol{x}})|do(\boldsymbol{x}_{\mathcal{S}} = \boldsymbol{x}_{\mathcal{S}})] - \mathbb{E}[\epsilon] = \mathbb{E}[g(\tilde{\boldsymbol{x}}, \overline{\boldsymbol{x}})|do(\boldsymbol{x}_{\mathcal{S} \cup \{i\}} = \boldsymbol{x}_{\mathcal{S} \cup \{i\}})] - \mathbb{E}[g(\tilde{\boldsymbol{x}}, \overline{\boldsymbol{x}})|do(\boldsymbol{x}_{\mathcal{S}} = \boldsymbol{x}_{\mathcal{S}})]$. Thus, the attribution are identical for models $f = g(\tilde{\boldsymbol{x}}, \overline{\boldsymbol{x}})$ and $f = g(\tilde{\boldsymbol{x}}, \overline{\boldsymbol{x}}) + \mathbb{E}[\epsilon]$ in intervention Shapley.

The derivative $\frac{\partial f}{\partial \boldsymbol{x}} = \frac{\partial g}{\partial \boldsymbol{x}}$ holds when model $f = g(\boldsymbol{x}) + \mathbb{E}[\epsilon]$ as $\mathbb{E}[\epsilon]$ is a constant. Thus, the attribution are identical for models $f = g(\tilde{\boldsymbol{x}}, \overline{\boldsymbol{x}})$ and $f = g(\tilde{\boldsymbol{x}}, \overline{\boldsymbol{x}}) + \mathbb{E}[\epsilon]$ in Integrated Gradients(IG). $\qquad \square$

# D Discrete Confounded features and Gradient-Free Model Training

## D.1 Training with Discrete Features

We extend our discussion to scenarios where the targets predicted by $\hat{M}_\phi (\tilde{\boldsymbol{x}} \mid \overline{\boldsymbol{x}}_t, \psi_t)$ are discrete. In cases where the prediction features P of $\hat{M}_\phi$ are discrete, the fundamental approach to optimizing the loss function $L(T; \theta)$ remains similar. The primary modification involves substituting the integral over the probability distribution of P with a summation across discrete points. Assuming P has K

categories, and denoting the probability of the $t^{th}$ data point being classified into the $k^{th}$ category by $\hat{M}_\phi\left(\tilde{\boldsymbol{x}}^k \mid \overline{\boldsymbol{x}}_t, z_t\right)$, the loss function when y is continuous is reformulated as:

$$\mathcal{L}(T; \theta) = |T|^{-1} \sum_t \left( y_t - \sum_{k=1}^K \hat{M}_\phi\left(\tilde{\boldsymbol{x}}^k \mid \overline{\boldsymbol{x}}_t, \psi_t\right) f_\theta\left(\tilde{\boldsymbol{x}}^k, \overline{\boldsymbol{x}}_t\right) \right)^2. \tag{7}$$

For the $t^{th}$ training data point, the gradient of this loss function is:

$$\nabla_\theta \mathcal{L}_t = -2 \left[ y_t - \sum_{k=1}^K \hat{M}_\phi\left(\tilde{\boldsymbol{x}}^k \mid \overline{\boldsymbol{x}}_t, \psi_t\right) f_\theta\left(\tilde{\boldsymbol{x}}^k, \overline{\boldsymbol{x}}_t\right) \right] \cdot \left[ \sum_{k=1}^K \hat{M}_\phi\left(\tilde{\boldsymbol{x}}^k \mid \overline{\boldsymbol{x}}_t, \psi_t\right) f_\theta'\left(\tilde{\boldsymbol{x}}^k \overline{\boldsymbol{x}}_t\right) \right]. \tag{8}$$

Furthermore, the gradients of a mini-batch comprising m training data tuples are computed as:

$$\nabla_\theta^m \mathcal{L}_t \equiv m^{-1} \sum_t -2 \left[ \left( y_t - \sum_{k=1}^K \hat{M}_\phi\left(\tilde{\boldsymbol{x}}^k \mid \overline{\boldsymbol{x}}_t, \psi_t\right) f_\theta\left(\tilde{\boldsymbol{x}}^k, \overline{\boldsymbol{x}}_t\right) \right) \right] \cdot \left[ \sum_{k=1}^K \hat{M}_\phi\left(\tilde{\boldsymbol{x}}^k \mid \overline{\boldsymbol{x}}_t, \psi_t\right) f_\theta'\left(\tilde{\boldsymbol{x}}^k, \overline{\boldsymbol{x}}_t\right) \right]. \tag{9}$$

In situations where y is a discrete variable, representing categories or classes, the multi-class cross-entropy can be formulated as:

$$\mathcal{L}(T; \theta) = |T|^{-1} \sum_t \sum_r^R \sum_{k=1}^K \hat{M}_\phi\left(\tilde{\boldsymbol{x}}^k \mid \overline{\boldsymbol{x}}_t, \psi_t\right) y_{t,r} \cdot \ln f_{\theta,r}\left(\tilde{\boldsymbol{x}}^k, \overline{\boldsymbol{x}}_t\right) \tag{10}$$

In this formulation, $y_{t,r}$ represents the true label of the $t^{th}$ data point in the $r^{\text{th}}$ category. $R$ denotes the total number of distinct categories or classes into which the target variable $y$ can be classified. The model's prediction for this category is given by $f_{\theta,r}\left(\tilde{\boldsymbol{x}}, \overline{\boldsymbol{x}}_t\right)$. The gradient calculation for the $t^{th}$ training data point, considering this loss function, is then:

$$\nabla_\theta \mathcal{L}_t = \sum_{r=1}^R \sum_{k=1}^K \hat{M}_\phi\left(\tilde{\boldsymbol{x}}^k \mid \overline{\boldsymbol{x}}_t, \psi_t\right) \frac{y_{t,r}}{f_{\theta,r}\left(\tilde{\boldsymbol{x}}^k, \overline{\boldsymbol{x}}_t\right)} \cdot f_{\theta,r}'\left(\tilde{\boldsymbol{x}}^k, \overline{\boldsymbol{x}}_t\right). \tag{11}$$

Furthermore, the gradients of a mini-batch comprising m training data tuples are computed as:

$$\nabla_\theta^m \mathcal{L}_t \equiv m^{-1} \sum_t \sum_{r=1}^R \sum_{k=1}^K \hat{M}_\phi\left(\tilde{\boldsymbol{x}}^k \mid \overline{\boldsymbol{x}}_t, \psi_t\right) \frac{y_{t,r}}{f_{\theta,r}\left(\tilde{\boldsymbol{x}}^k, \overline{\boldsymbol{x}}_t\right)} \cdot f_{\theta,r}'\left(\tilde{\boldsymbol{x}}^k, \overline{\boldsymbol{x}}_t\right). \tag{12}$$

This adaptation of the loss function for discrete target variables ensures that our model can handle classification tasks, effectively optimizing its performance across multiple categories.

### D.2  Gradient-Free Model Training

When training models to fit $\hat{y}$ in scenarios where gradient-based optimization is not feasible, we introduce an alternative approach that effectively addresses the influence of confounding factors. The essence of this approach lies in the generation of synthetic data, which is derived from the predicted distribution of p. By sampling each original data point B times, we create B synthetic data points for every original point. This process results in a synthetic dataset that embodies the controlled effects of the confounders. The creation of this dataset is a vital step towards ensuring that the subsequent model training is less influenced by confounding variables.

A key advantage of this method is its independence from any specific model type. The generated synthetic dataset can be utilized to train a variety of machine learning models, not limited to those that rely on gradient-based optimization. This model-agnostic nature significantly widens the applicability of our approach, making it suitable for various scenarios and models. Through this method, we ensure that the training of models occurs in an environment where the impact of confounders is mitigated, thereby enhancing the reliability of the feature attribution.

# E   Supplement to SHAP and IG approximation

SHAP-based methods and IG-based methods can be applied to the proposed confounder-free models. However, the computational complexity poses a significant barrier to real-world applications. The exact computation of the Shapley value is proved to be an #P-hard problem [11], and the exact computation of integrated gradients requires the antiderivative of the gradient, which is infeasible due to their complexity, necessitating the use of approximation methods. The approximation cost of the Shapley value is higher than the straightforward sampling in the path of integrated gradients due to extensive feature subset evaluations. To further enhance the applicability, we develop optimizations for the approximation of SHAP-based methods. Research has shown that Shapley values can be represented not only based on marginal contributions but also complementary contributions, which allows for reusing samples in estimations, offering an advantage [46]. We propose an enhanced approach for SHAP-based methods, optimizing complementary contribution-based sampling using confidence intervals. This optimization is designed to minimize estimation errors in all utility functions within SHAP-based methods.

## E.1   Estimation Techniques for SHAP

Recent work [46] suggests that the Shapley value formula can be equivalently transformed into a form expressed based on complementary contributions. Here, the complementary contribution refers to the difference in utility between complementary subsets. The Shapley expression is given by

$$\mathcal{SV}_i = \frac{1}{|\mathcal{N}|} \sum_{\mathcal{S} \subseteq \mathcal{N} \setminus \{i\}} \frac{\mathcal{U}(\mathcal{S} \cup \{i\}) - \mathcal{U}(\mathcal{S})}{\binom{|\mathcal{N}|-1}{|\mathcal{S}|}} \tag{13}$$

$$= \frac{1}{n} \sum_{\mathcal{S} \subseteq \mathcal{N} \setminus \{z_i\}} \frac{\mathcal{U}(\mathcal{S} \cup \{z_i\}) - \mathcal{U}(\mathcal{N} \setminus (\mathcal{S} \cup \{z_i\}))}{\binom{n-1}{|\mathcal{S}|}}. \tag{14}$$

The formulas based on complementary contributions offer advantages in terms of sample reusability during approximate computation. Building on this foundation, we propose a dynamic sampling adjustment based on the confidence intervals of Shapley value estimates in stratified sampling.

Denote by $\mathfrak{S}_{\mathcal{N}}^{i,j} = \{\mathcal{S} \cup \{z_i\} | \mathcal{S} \subseteq \mathcal{N} \setminus \{z_i\}, |\mathcal{S}| = j - 1\}$ $(1 \leq j \leq n)$ the set of $(z_i, j)$-coalitions, and by $\mathcal{SV}_{i,j}$ the expected complementary contributions of $(z_i, j)$-coalitions. That is,

$$\mathcal{SV}_{i,j} = \sum_{\mathcal{S} \in \mathfrak{S}_{\mathcal{N}}^{i,j}} \frac{\mathcal{U}(\mathcal{S}) - \mathcal{U}(N \setminus \mathcal{S})}{\binom{n-1}{j-1}}. \tag{15}$$

Complementary contributions $CC(\mathcal{S}) = \mathcal{U}(\mathcal{S} \cup \{z_i\}) - \mathcal{U}(\mathcal{N} \setminus (\mathcal{S} \cup \{z_i\}))$ are naturally stratified into $n$ strata $\mathfrak{S}_{\mathcal{N}}^{i,1}, \ldots, \mathfrak{S}_{\mathcal{N}}^{i,n}$ according to the coalition size. We start by deriving the confidence interval of the estimator of $\mathcal{SV}_{i,j}$ using $t$-test.

**Lemma 5.** *According to the Central Limit Theorem, the sample mean approximates a normal distribution when the sample size is sufficiently large. Assuming a confidence level of $\alpha$, the confidence interval for $\overline{\mathcal{SV}_{i,j}}$ based on the t-test is $\overline{\mathcal{SV}_{i,j}} \pm A_\alpha \frac{S_{i,j}}{\sqrt{m_{i,j}}}$ , which can also be represented as*

$$P(\overline{\mathcal{SV}_{i,j}} - A_\alpha \frac{S_{i,j}}{\sqrt{m_{i,j}}} < \mathcal{SV}_{i,j} < \overline{\mathcal{SV}_{i,j}} + A_\alpha \frac{S_{i,j}}{\sqrt{m_{i,j}}}) = \alpha, \tag{16}$$

*where $A_\alpha$ is the t-score corresponding to $\alpha$, $m_{i,j}$ is the sample size of $\mathfrak{S}_{\mathcal{N}}^{i,j}$ and $S_{i,j}$ is the sampling variance of $\overline{\mathcal{SV}_{i,j}}$.*

Denote by $\mathfrak{S}_{\mathcal{N}}^j = \{\mathcal{S} | \mathcal{S} \subseteq \mathcal{N}, |\mathcal{S}| = j\}$ the set of $j$-coalitions $(1 \leq j \leq n)$. After drawing a coalition $\mathcal{S}$ from $\mathfrak{S}_{\mathcal{N}}^j$, we can estimate the complementary contribution $CC_{\mathcal{N}}(\mathcal{S})$, which can be used in $\mathcal{SV}_{i,j}$ for $z_i$ in $\mathcal{S}$ and $\mathcal{SV}_{i,n-j}$ for $z_i$ in $\mathcal{N} \setminus \mathcal{S}$. In light of the fact that each sample can influence multiple strata, how should we allocate the number of samples to optimize the precision of the estimated values? We denote $I_{\mathcal{N}}^j$ as the sum of the confidence intervals for strata which can be influenced by a random sample of $\mathfrak{S}_{\mathcal{N}}^j$, that is

$$I_{\mathcal{N}}^j = 2 * (\sum_{i=1}^N A_\alpha \frac{S_{i,j}}{\sqrt{m_{i,j}}} + \sum_{i=1}^N A_\alpha \frac{S_{i,n-j}}{\sqrt{m_{i,n-j}}}). \tag{17}$$

For each sampling iteration, we select the stratum that maximizes the sum of the corresponding confidence intervals.

Next, we will outline the algorithmic procedure, which is divided into two primary stages. Due to page limitations, the pseudo-code of the algorithm is presented in Algorithm 1 in the appendix. In the first stage, we sample at least $m_{\text{init}}$ samples for $\mathcal{SV}_{i,j}$. We then compute unbiased estimations of $\sigma_{i,j}^2$ using Bessel's correction based on samples collected in the first stage. In the second stage, the stratum for each individual sampling is determined based on the sum of confidence intervals $I_{\mathcal{N}}^j(1 \leq j \leq n/2)$. Furthermore, this sum is updated upon the completion of each sampling. Specifically, let $CC_{\mathcal{N}}(\mathcal{S}_1 \cup \{z_i\}), \ldots, CC_{\mathcal{N}}(\mathcal{S}_{m_{i,j}} \cup \{z_i\})$ be $m_{i,j}$ samples for computing $\overline{\mathcal{SV}_{i,j}}$, then $\widehat{\sigma_{i,j}^2} = \frac{1}{m_{i,j}-1} \sum_{k=1}^{m_{i,j}} (CC_{\mathcal{N}}(\mathcal{S}_k \cup \{z_i\}) - \frac{1}{m_{i,j}} \sum_{k=1}^{m_{i,j}} CC_{\mathcal{N}}(\mathcal{S}_k \cup \{z_i\}))$. Let $m_{\text{first}}$ be the number of samples used in the first stage, and the number of remaining samples is $m - m_{\text{first}}$. We calculate $I_{\mathcal{N}}^j(1 \leq j \leq n/2)$ according to Equation (17) using the unbiased sample variance $\widehat{\sigma_{i,j}^2}$. We randomly sample from the stratum with the largest sum of confidence intervals. Following this sampling, the sum of confidence intervals will be updated. We will continue to repeat this process until all samples have been utilized. The final estimation of Shapley value is the average of all complementary contribution means in each stratum.

SHAP experiences an exponential increase in computational complexity with the addition of more features. However, for IG, the increase in features does not significantly escalate the complexity of the integral path. Therefore, in terms of efficiency in approximate computations, IG generally outperforms SHAP. This makes IG a preferable choice in situations where computational complexity for interpretability is a critical concern. However, it is important to note that SHAP, as a model-agnostic method, is applicable for interpreting models that are not based on gradient optimization.

**Algorithm of SHAP Computation Based on the Confidence Interval.** For the algorithm process we propose, which utilizes confidence intervals to optimize the calculation of Shapley values, refer to Algorithm 1. It is important to note that our algorithm is utility function-agnostic [29], meaning it can be applied across various SHAP-based algorithm variants. This is achieved by simply substituting the utility function defined by each method into our calculation. Furthermore, we provide an unbiased proof of our method in Theorem 6.

**Theorem 6.** *Given a set of players $\mathcal{N} = \{z_1, \ldots, z_n\}$, Algorithm 1 gives an unbiased estimation of Shapley value for every player, that is, $E[\overline{\mathcal{SV}_i}] = \mathcal{SV}_i$ $(1 \leq i \leq n)$.*

*Proof.* Denote by $CC_{\mathcal{N}}(\mathcal{S}_1), \ldots, CC_{\mathcal{N}}(\mathcal{S}_{m_{i,j}})$ a sample of $\mathfrak{S}_{\mathcal{N}}^{i,j}$ $(1 \leq i,j \leq n)$ drawn by Algorithm 1. The expectation of the sample $\overline{\mathcal{SV}_{i,j}} = \frac{1}{m_{i,j}} \sum_{k=1}^{m_{i,j}} CC_{\mathcal{N}}(\mathcal{S}_k)$. We can compute the expectation of $\overline{\mathcal{SV}_{i,j}}$ with

$$E[\overline{\mathcal{SV}_{i,j}}] = E[\frac{1}{m_{i,j}} \sum_{k=1}^{m_{i,j}} CC_{\mathcal{N}}(\mathcal{S}_k)] = \frac{1}{m_{i,j}} \sum_{k=1}^{m_{i,j}} E[CC_{\mathcal{N}}(\mathcal{S}_k)] \tag{18}$$

According to Equation 15, $E[CC_{\mathcal{N}}(\mathcal{S}_k)] = \mathcal{SV}_{i,j}$. Thus, $E[\overline{\mathcal{SV}_{i,j}}] = \mathcal{SV}_{i,j}$ that means $\overline{\mathcal{SV}_{i,j}}$ is an unbiased estimation of $\mathcal{SV}_{i,j}$.

Then, we can compute the expectation of $\overline{\mathcal{SV}_i}$ produced by Algorithm 1. We have

$$E[\overline{\mathcal{SV}_i}] = E[\frac{1}{n} \sum_{j=1}^{n} \overline{\mathcal{SV}_{i,j}}] = \frac{1}{n} \sum_{j=1}^{n} E[\overline{\mathcal{SV}_{i,j}}] = \frac{1}{n} \sum_{j=1}^{n} \mathcal{SV}_{i,j} = \mathcal{SV}_i. \tag{19}$$

That is, $\overline{\mathcal{SV}_i}$ is an unbiased estimation of $\mathcal{SV}_i$. $\qquad\square$

### E.2    Unbiased Integrated Gradients Approximation

In existing literature related to Integrated Gradients (IG), interpolation methods are commonly used for approximation [13], which already exhibit high efficiency compared to SHAP-like approximation methods. In contrast, our paper introduces the use of Monte Carlo methods for the integration of gradients. It's important to clarify that the aim of proposing an unbiased estimate for IG is not

**Algorithm 1** Shapley value computation based on the confidence interval.

---

**Input:** players $\mathcal{N} = \{z_1, \ldots, z_n\}$, $m_{init} > 1$ , and $m > 0$
**Output:** approximate Shapley value $\overline{\mathcal{SV}}_i$ for each player $z_i$ $(1 \leq i \leq n)$
$\overline{\mathcal{SV}}_i, \overline{\mathcal{SV}}_{i,j}, m_{i,j} \leftarrow 0$ $(1 \leq i \leq n)$;
$c \leftarrow -1$;
**while** $c \neq \sum_{j=1}^{n} m_{1,j}$ **do**
  $c = \sum_{j=1}^{n} m_{1,j}$;
  **for** i=1 to n, j = 1 to n **do**
    **if** $m_{i,j} < m_{init}$ **then**
      let $\mathcal{S}$ be a sample drawn from $\mathfrak{S}_{\mathcal{N}}^{j}$;
      $u \leftarrow \mathcal{U}(\mathcal{S}) - \mathcal{U}(\mathcal{N} \setminus \mathcal{S})$;
      **for** $z_i \in \mathcal{S}$ **do**
        $\overline{\mathcal{SV}}_{i,|\mathcal{S}|} += u$; $m_{i,|\mathcal{S}|} += 1$;
      **end for**
      **for** $z_i \in \mathcal{N} \setminus \mathcal{S}$ **do**
        $\overline{\mathcal{SV}}_{i,|\mathcal{N}\setminus\mathcal{S}|} -= u$; $m_{i,|\mathcal{N}\setminus\mathcal{S}|} += 1$;
      **end for**
    **end if**
  **end for**
**end while**
compute $\widehat{S_{i,j}^2}$ $(1 \leq i, j \leq n)$;
$m_{first} \leftarrow \sum_{j=1}^{n} m_{1,j}$;
**for** k = 0 to $m - m_{first}$ **do**
  **for** j=1 to n **do**
    $I_{\mathcal{N}}^{j} = 2 * (\sum_{i=1}^{N} A_\alpha \frac{S_{i,j}}{\sqrt{m_{i,j}}} + \sum_{i=1}^{N} A_\alpha \frac{S_{i,n-j}}{\sqrt{m_{i,n-j}}})$;
  **end for**
  let $\mathcal{S}$ be a sample drawn from $\mathfrak{S}_{\mathcal{N}}^{j}$ where j corresponding to the stratum with the maximum $I_{\mathcal{N}}^{j}$;
  $u \leftarrow \mathcal{U}(\mathcal{S}) - \mathcal{U}(\mathcal{N} \setminus \mathcal{S})$;
  **for** $z_i \in \mathcal{S}$ **do**
    $\overline{\mathcal{SV}}_{i,|\mathcal{S}|} += u$; $m_{i,|\mathcal{S}|} += 1$;
  **end for**
  **for** $z_i \in \mathcal{N} \setminus \mathcal{S}$ **do**
    $\overline{\mathcal{SV}}_{i,|\mathcal{N}\setminus\mathcal{S}|} -= u$; $m_{i,|\mathcal{N}\setminus\mathcal{S}|} += 1$;
  **end for**
  update $\widehat{S_{i,j}^2}$ $(1 \leq i, j \leq n)$;
**end for**
**for** i=1 to n **do**
  $\overline{\mathcal{SV}}_i = \frac{1}{n} \sum_{j=1}^{n} \overline{\mathcal{SV}}_{i,j}/m_{i,j}$;
**end for**
**return** $\overline{\mathcal{SV}}_1, \ldots, \overline{\mathcal{SV}}_n$.

---

to optimize sampling efficiency but rather to provide an error analysis for the sampling process. Our proposed approach not only provides an unbiased estimate of the integrated gradients but also includes this crucial error analysis.

Let u be a uniformly distributed random variable over the interval $[0, 1]$, where

$$h(u) = (x_i - x_i')\frac{\partial f(x' + u(x - x'))}{\partial x_i}.$$

Then, h(u) is an unbiased estimator of $IG_i(x, x', f)$ due to

$$E[h(u)] = (x_i - x_i')\int_{u=0}^{1}\frac{\partial f(x' + u(x - x'))}{\partial x_i}du = (x_i - x_i')\int_{\alpha=0}^{1}\frac{\partial f(x' + \alpha(x - x'))}{\partial x_i}d\alpha.$$

(20)

Next, we can obtain a more accurate unbiased estimate through the following steps: First, we generate $m_i$ random samples $u_1, \cdots, u_{m_i}$ of u. Then, we sample h(u) based on these random numbers to get an independent and identically distributed sample $h(u_1), \cdots, h(u_{m_i})$. Finally, we use the observed values of $\overline{IG_i(x, x', f)} = \frac{1}{m_i}\sum_{i=1}^{m_i}h(u_i)$ as the estimate for $IG_i(x, x', f)$. The proof that $\frac{1}{m_i}\sum_{i=1}^{m_i}h(u_i)$ is an unbiased estimator of $IG_i(x, x', f)$ is straightforward, due to the fact that $E[\frac{1}{m_i}\sum_{i=1}^{m_i}h(u_i)] = \frac{1}{m_i}\sum_{i=1}^{m_i}E[h(u_i)] = IG_i(x, x', f)$.

**Lemma 7.** *The probability that $\overline{IG_i(x, x', f)}(1 \leq i \leq n)$ deviates from $IG_i(x, x', f)$ be equal to or greater than any fixed $\epsilon \geq 0$ given the sample size $m_i$ is bounded by*

$$\mathbb{P}(|\overline{IG_i(x, x', f)} - IG_i(x, x', f)| \geq \epsilon | m_i) \leq 2\exp(-\frac{2m_i\epsilon^2}{r_{i,j}^2})$$

(21)

*where $r_{i,j} = \max_{u \in [0,1]} h(u) - \min_{u \in [0,1]} h(u)$.*

*Proof.* According to Hoeffding's inequality, we have

$$\mathbb{P}(|\overline{IG_i(x, x', f)} - IG_i(x, x', f)| \geq \epsilon | m_i)$$

(22)

$$= \mathbb{P}(|\overline{IG_i(x, x', f)} - \mathbb{E}[\overline{IG_i(x, x', f)}]| \geq \epsilon | m_i)$$

(23)

$$= \mathbb{P}(|\sum_{i=1}^{m_i} h(u_i) - \mathbb{E}[\sum_{i=1}^{m_i} h(u_i)]| \geq m_i\epsilon | m_i) \leq 2\exp(-\frac{2m_i\epsilon^2}{r_{i,j}^2}).$$

(24)

$\square$

# F    Supplement to Experiments

## F.1    Experiments Compute Resources

We conduct experiments on a machine with 2 Montage(R) Jintide(R) C6226R @ 2.90GHz and 256GB memory. Our experiments do not require high-end hardware, and our algorithm is not time-consuming. For feature attribution experiments on synthetic datasets, each attribution algorithm takes about 10 seconds to attribute one data point. Thus, an attribution algorithm takes several hours to attribute an entire dataset. The time consumption on our real dataset is similar. Also, since our algorithm uses very little memory, it is easy to run multiple processes and algorithms in parallel on a machine, so the time cost for reproduction is friendly.

## F.2    Analysis of Experiment Design

**Statistical Characteristics of the Synthetic Datasets.** We present the mean and variance of each feature in our synthetic datasets to provide crucial insights into their characteristics, as shown in Table 3. The mean offers an understanding of the average behaviour of features, while the variance indicates their variability. This information is vital for assessing the data's overall distribution and quality, and it plays a key role in interpreting the results of our proposed methods and baseline algorithms.

Table 3: Mean and Std.Dev. of Features as a Function of $\rho$.

| $\rho$ | 0.2 | 0.4 | 0.6 | 0.8 | 1.0 |
|--------|-----|-----|-----|-----|-----|
| $e_a$ | $0.10 \pm 0.05$ | $0.20 \pm 0.11$ | $0.30 \pm 0.17$ | $0.40 \pm 0.23$ | $0.50 \pm 0.28$ |
| $t_a$ | $0.21 \pm 0.12$ | $0.27 \pm 0.14$ | $0.32 \pm 0.17$ | $0.37 \pm 0.18$ | $0.42 \pm 0.21$ |
| $y_a$ | $0.68 \pm 0.18$ | $0.87 \pm 0.24$ | $1.0 \pm 0.31$ | $1.15 \pm 0.39$ | $1.30 \pm 0.47$ |
| $e_b$ | $2.03 \pm 0.12$ | $1.12 \pm 0.13$ | $0.83 \pm 0.15$ | $0.69 \pm 0.16$ | $0.61 \pm 0.18$ |
| $t_b$ | $1.11 \pm 0.21$ | $0.72 \pm 0.19$ | $0.60 \pm 0.18$ | $0.53 \pm 0.18$ | $0.50 \pm 0.18$ |
| $y_b$ | $2.83 \pm 0.25$ | $1.65 \pm 0.23$ | $1.27 \pm 0.24$ | $1.09 \pm 0.25$ | $0.99 \pm 0.27$ |

These statistics help validate the synthetic data's consistency and reliability, which is essential for the credibility of our experimental findings.

**Statistical Characteristics of the Real Datasets.** In the analysis of the Griliches76 dataset [36], we observe various degrees of correlation between the mother's years of education (denoted as med) and several key variables. Firstly, the correlation coefficient between the mother's education and marital status (variables mrt and mrt80) is close to 0, indicating almost no correlation between the mother's level of education and her marital status. The correlation coefficients with urban residence status (variables smsa and smsa80) are 0.098 and 0.031, respectively, suggesting a slight positive correlation. This implies that there is a weak but positive association between the mother's educational attainment and living in an urban area. A moderate positive correlation is observed with IQ, as indicated by a correlation coefficient of 0.226 with the mother's education. This suggests that higher maternal education is somewhat associated with higher IQ scores. Similarly, the correlation between the mother's education and scores in the world of work test is 0.195, which also reflects a moderate positive correlation. This indicates that higher maternal education levels might be linked to better performance in job-related knowledge. Most notably, the correlation coefficients with personal education years (variables s and s80) are 0.340 and 0.341, respectively, indicating a relatively strong positive correlation. This suggests that the mother's level of education is considerably associated with the individual's own educational attainment. While there is a certain degree of correlation between maternal education and both IQ and job knowledge test scores, factors like regression to the mean in intelligence suggest that the correlation between a mother's education and her child's IQ is weaker than the correlation between a mother's education and the child's own educational attainment. Therefore, we posit that selecting maternal education as an instrumental variable, although not perfectly ideal, still holds validity and can be utilized to verify our methodology.

In the Angrist dataset, a child must be six years old within the current year to enroll in school under the U.S. Compulsory Education Law. In the U.S., the school year typically starts in August, meaning a child turning six in December can still commence their education in the same year. Consequently, a child born in the fourth quarter, such as December, can start school before reaching six. Conversely, a child born in the first quarter, like January, must wait until the autumn term after their sixth birthday to begin school. U.S. law mandates students must be at least 16 years old to legally drop out of school. Therefore, students dropping out at 16 may have varying years of education based on their birth month. For instance, those born between 1920-1929 have average educational years of 11.39, 11.44, 11.55, and 11.57 for each quarter, respectively. Parents, when deciding to have children, seldom consider such subtle differences in birth months. Thus, the month of a child's birth, independent of other factors affecting educational levels like intelligence, family background, and environment, can be seen as a random assignment. This inadvertently creates variations in education duration based on birth month – akin to a randomized controlled trial where children born in the fourth quarter represent the "experimental group" with longer education, while those in the first quarter are the "control group" with shorter education. Hence, the birth quarter serves as an instrumental variable in this context.

### F.3 Classification Task with DNN Model

In this experiment, we continued to utilize synthetic datasets a and b for a classification study. This time, the labels were processed for binary classification. Specifically, we computed the probabilities for data points being classified into category 1 by applying a sigmoid function to the y values in the datasets; otherwise, the labels were assigned to category 0. Due to the inherent randomness in

generating labels, it was not feasible to directly determine a benchmark for each feature's contribution to the data classification.

To validate the efficacy of our proposed method, we designed a comparative experiment based on the symmetric properties of SHAP and IG. In this experiment, we set baseline inputs by reducing the values of feature t and the collaborative variables c in each data point. The reduction followed a specific rule: the decrease in t and the decrease in c should result in equivalent Shapley/IG values for the change in y. Under this setup, the SHAP/IG values attributed to the classification into category 1 should be identical for both t and c. We assessed the effectiveness of our approach by comparing the difference in SHAP/IG values for t and c between our method and baseline algorithms. The results indicated that our approach significantly reduced errors compared to baseline algorithms. This outcome suggests that even though $t$ and $c$ might have similar Shapley values in altering y, the baseline training method may inaccurately estimate changes in latent confounders, leading to different impacts of $t$ and $c$ on the final classification.

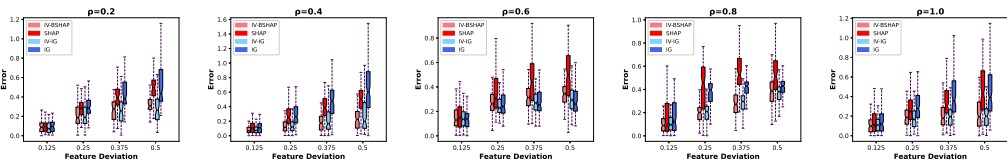

Figure 4: Evaluation results on synthetic Dataset A with DNN Classifier.

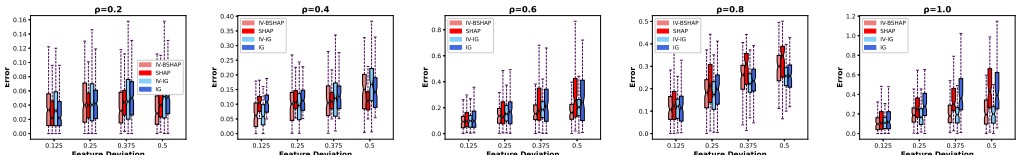

Figure 5: Evaluation results on synthetic Dataset B with DNN Classifier.

## F.4    Regression Task with XGBoost Model

We opted for XGBoost as the representative of non-deep learning models for our experiments. As gradient accumulation is not feasible on XGBoost, the Integrated Gradients (IG) algorithm cannot be applied. Hence, our comparisons were primarily focused on SHAP-based algorithms. The experimental results indicate that our method outperforms the baseline in most scenarios. When the impact of the unobservable confounder is minimal, our method is less effective compared to the baseline. This is considered reasonable, as there are inherent errors in training the model with features re-estimated using instrumental variables. In such scenarios, the influence of these errors on the model surpasses that of the unobservable confounder.

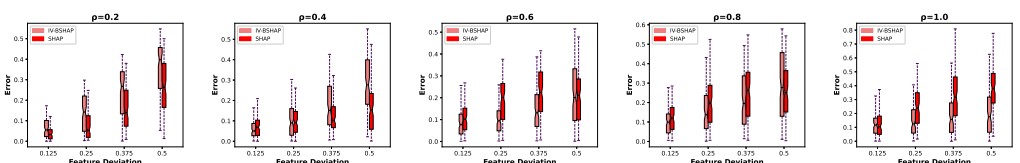

Figure 6: Evaluation results on synthetic dataset a with XGBoost.

## F.5    Efficiency of Our Approximation Methods

We utilized widely-used algorithms MC (Monte Carlo, referring to the Monte Carlo sampling method based on marginal contributions) [7], CC (Complementary Contribution, referring to the stratified sampling method based on complementary contribution), and the state-of-the-art CCN

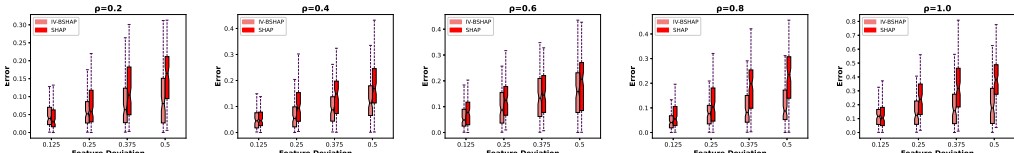

Figure 7: Evaluation results on synthetic dataset b with XGBoost.

Table 4: MSE of the SHAP values estimation.

| SAMPLES | 56*100 | 56*200 | 56*300 | 56*400 | 56*500 |
|---|---|---|---|---|---|
| MC | 3.20E-5 | 1.74E-5 | 1.28E-5 | 9.81E-6 | 8.37E-6 |
| CC | 1.78E-5 | 9.45E-6 | 6.89E-6 | 5.37E-6 | 4.58E-6 |
| CCN | 1.76E-5 | 9.62E-6 | 6.71E-6 | 5.45E-6 | 4.96E-6 |
| **OURS** | **1.48E-5** | **8.66E-6** | **6.15E-6** | **4.92E-6** | **4.25E-6** |

(Complementary Contribution Neyman, referring to the sampling based on Neyman allocation with complementary contribution) [46] as baseline methods for our experiment on the real-world Spambase dataset. We chose the Spambase dataset for its larger number of features (56), compared to the two other real datasets we previously used. This higher feature count offers a better testbed to evaluate our proposed methods. It is worth noting that both our proposed method and these baseline methods are unbiased sampling estimation approaches. We trained a regression neural network model on a random selection of 1000 data points from this dataset. The efficiency of each method was assessed by comparing the Mean Squared Error (MSE) of SHAP values against a benchmark for the same number of samples. For this, we denoted the SHAP value of the $j^{th}$ feature for the $i^{th}$ data point as $\mathcal{SV}_{i,j}$ in the benchmark, and $\overline{\mathcal{SV}_{i,j}}$ in the estimation algorithm, with the MSE calculated using $\frac{\sum_{i=1}^{n}\sum_{j=1}^{m}(\mathcal{SV}_{i,j}-\overline{\mathcal{SV}_{i,j}})^2}{n*m}$, where n=1000 and m=56. Given that the computation of exact SHAP values requires exponential time complexity, we used the results obtained from extensive sampling via the CC method as our benchmark, involving 10,000×56 samples. These results served as a specific benchmark, separate from the baseline CC method used earlier in the experiment for comparative analysis. We then analyzed errors for sampling algorithms at various sample sizes, ranging from 100×56 to 500×56. Our findings revealed a decrease in error for all algorithms as sample size increased, with our method exhibiting the lowest error as shown in Table 4. This superior performance is attributed to our method's unique ability to estimate each stratum's confidence interval from sample variance during sampling.

