# OpenReview forum: "Data-faithful Feature Attribution: Mitigating Unobservable Confounders via Instrumental Variables"
_NeurIPS.cc/2024/Conference — NeurIPS 2024 poster_

### Official Review · Reviewer_6T6P · 2024-06-19

**Soundness:** 3
**Presentation:** 3
**Contribution:** 3
**Rating:** 7
**Confidence:** 3

**Summary:**

The paper introduces the problem that unobservable confounders in data can mislead and negatively affect the quality of feature attribution explanations. Feature attribution methods do not consider unobservable confounders which poses the risk of misinterpreting the attribution scores found, which could have consequences, especially in high-risk application domains where users rely on feature attribution to calibrate their trust in the model. To address the issue, the authors propose to decouple input features affected by unobservable confounding factors and those not. The latter are then used to train a confounder-free model, for which they show that feature attribution methods SHAP and Integrated Gradients are faithful to the true data generation process and robust.

**Strengths:**

- The paper addresses a novel perspective on feature attribution: Unobserved confounders in the data generation process can mislead feature attribution methods to assign inflated or deflated scores to features that are affected by unobserved confounders. I have not come across this perspective of connecting the data generation process to local explanations before. I believe it is an interesting viewpoint, particularly for tabular data use cases where causal relations may be discoverable.
- The paper uses an example to illustrate the problem it is addressing and the approach to solving it, which helps the reader follow the methodology.
- The authors provide the source code for their experiments.

**Weaknesses:**

- The paper is quite inline-math-text heavy, which limits its readability. I believe the readability and therefore clarity can be improved by restructuring math-heavy paragraphs (room for this could be made by e.g. shortening the introduction).
- Some things were unclear, please see the questions.
- Minor comments:
     - Lines 146-154: The long equations are slightly difficult to follow when written inline, and will be clearer if using \equation or \align.
     - Lines 156-159: Thought experiment results could be shown in a table, for more clarity.
     - Line 191: it’s -> it is
     - Please increase the font size in Figures 2 and 3 for better readability.

**Questions:**

- I did not fully understand how you identified the unobservable confounders in the real dataset use case. How could practitioners identify unobservable confounders in their models?
- Feature attribution methods explain a model output $f(x)$ by assigning attribution scores in the input instance $x$. I am unsure if I may have misunderstood your approach, but by fitting a model with a different data (one time with all variables, and once without variables affected by unobserved confounders), are you not training and explaining a different model? If this is the case, I see your contribution rather providing a training procedure on only instrumental variables and without confounding variables, leading to data-faithful attribution. Currently, the paper's contribution often reads as if you are proposing a new attribution method (i.e. referring to your approaches as IV-SHAP and IV-IG). Could you please clarify?
- By excluding variables affected by unobserved confounders in training, does the model's predictive performance (e.g. measured in accuracy) suffer or does the task become easier for the model? Is there a trade-off between data-faithful feature attribution and model performance?
- How similar are the resulting attribution scores of IV-SHAP and IV-IG? It has previously been found that local explanation methods like feature attribution often disagree because they only locally approximate the model (Krishna et al., 2022: The Disagreement Problem in Explainable Machine Learning). Does removing the confounding variables lead to less disagreeing feature attribution?
- In feature attribution, we are often interested in the features with the largest positive or negative attribution scores as we want to understand which features a model relied on most for its output. In this case, the faithful ranking of the scores matters more than the score itself. I'm curious, in your experiments, did the ranking of scores change with the instrumental variable approach? How much does being faithful to the data matter in practice if not?
- Out of curiosity: What do you think, how would this approach / does this approach transfer to high-dimensional problems like image or text classification?

I will happily adjust the score when I understand the paper better.

**Limitations:**

Limitations and broader impact are adequately discussed in the appendix.

---

> ### Author Rebuttal · Authors · 2024-08-06
>
> Thank you for your encouraging and insightful comments. Please find our response below.
>
> **Response to Weaknesses**
> * W1: We will follow your suggestions to enhance the readability.
>
> * Minor comments: Fixed. Thanks!
>
> **Response to Questions**
>
> * Q1: In our paper, when addressing the impact of unobserved confounders, it is not necessary to identify what these confounders are. Instead, we only need to determine their existence and, if present, employ our method to mitigate their effect on feature attribution. For example, we can use potential instrumental variables for detection. When we intervene on an instrumental variable, we can then analyze whether the correlation between variable $\tilde{\boldsymbol{x}}$ and the target feature aligns with the change of the target variable caused by the instrumental variable. If the correlation is strong, but the change in the target variable caused by the instrumental variable is small, this suggests the presence of unobserved confounders which increases the correlation between $\tilde{\boldsymbol{x}}$ and the target feature [1].
>
> * Q2: Feature attribution indeed involves explaining a model output $f(x)$ by assigning attribution scores to the input instance
> $x$. However, our focus is on data-faithful feature attribution, i.e., we are not trying to explain the output of a specific model but trying to explain the target feature $y$ through a model. For example, consider a medical scenario where a patient is interested in understanding the impact of all features on a disease, rather than just the prediction output of a particular diagnostic model. In data-faithful attribution, the trained model is used to estimate feature subset utilities because the real-world data generation equation is unknown. Thus, the model serves as a tool for data-faithful attribution.  We propose a training method to reduce confounder influence and enhance data-faithful attribution.  We will clarify them in the paper.
>
> * Q3: Our experiments suggest that the model's accuracy might be affected, indicating a trade-off in most scenarios. When we train the model using features that have been re-estimated to remove the influence of confounders, the model may lose the correlation between the predicted output and the confounders, which also affects the target feature $y$. We compared the Mean Squared Error (MSE) of our model with a predictive model in the experiment and found our loss to be slightly higher. We note that our goal is not to maximize the predictive accuracy of the model but rather to ensure that the attribution results are more faithful to data. We will clarify it in the paper.
>
> * Q4: In our experiments, the average change rates between IV-SHAP and IV-IG scores in both synthetic and real-world datasets are below 10\%. SHAP and IG are inspired by the Shapley and Aumann-Shapley values [2], respectively, and share many common properties [3]. In contrast, methods like LIME and DeepLIFT follow different principles and produce different values. Thus, merely removing confounders is unlikely to resolve inconsistencies between LIME/DeepLIFT and IV-SHAP/IV-IG. Discrepancies among attribution methods also arise from how they handle correlations between input features. For example, on-manifold Shapley [4] uses conditional expectations to supplement unselected features, while causal Shapley [5] uses interventions. These methods can be combined with ours, but addressing confounders alone cannot resolve differences due to varying approaches to feature correlation. Our paper focuses on the impact of unobserved confounders, and we suggest choosing the appropriate method for handling feature correlations based on the specific context.
>
> * Q5: The ranking of scores obtained from our methods and existing methods are different. We statistically analyze the proportion of attribution score ranking changes for IV-SHAP and IV-IG compared to SHAP and IG on Dataset A when $\rho=0.2$. The results are shown in the table below.
>
>      |     Feature Deviation     | 0.125   | 0.250   | 0.375   | 0.500   |
>      |----------|---------|---------|---------|---------|
>      | IV-SHAP  | 82.7%   | 87.1%   | 92.3%   | 94.2%   |
>      | IV-IG    | 58.4%   | 69.6%   | 77.1%   | 86.3%   |
>
>      In the Griliches76 dataset experiments, 57.1\% of the attribution rankings changed. Even when the rankings remain the same,
>      changes in attribution scores can offer valuable insights into feature contributions. For example, if the attribution score for
>      education decreases, even without a change in ranking, it could indicate that the return on education for income is lower than
>      initially expected.
> * Q6: The proposed feature attribution approach may be feasible for image and text classification. Images often encounter confounders like lighting. If a model is trained where most cat images are in strong lighting and dog images in weak lighting, the attribution for a cat image taken in weak lighting may be misleading. Identifying suitable instrumental variables to mitigate the confounders can make feature attribution more reliable and intuitive. This is an insightful extension, and we will explore it further.
>
> References
>
> [1] Baiocchi M, Cheng J, Small D S. Instrumental variable methods for causal inference[J]. Statistics in medicine, 2014.
>
> [2] Tauman Y. The Aumann-Shapley prices: a survey[J]. The shapley value, 1988.
>
> [3] Sundararajan M, Najmi A. The many Shapley values for model explanation[C]//(ICLR), 2020.
>
> [4] Frye C, de Mijolla D, Begley T, et al. Shapley explainability on the data manifold[C]//(ICLR), 2020.
>
> [5] Heskes T, Sijben E, Bucur I G, et al. Causal shapley values: Exploiting causal knowledge to explain individual predictions of complex models[J]. (NeurIPS), 2020.

---

> > ### Comment · Reviewer_6T6P · 2024-08-08
> >
> > Thank you for the clarifications and additional analysis for Q5. There are only a few follow-up questions and comments:
> >
> > Q1: Could you please confirm whether my understanding is correct? Unobservable confounders are not required to be found. By being able to observe their effects, it is possible to identify the instrumental variables and make use of them to train a model with data-faithful feature attributions.
> >
> > Q2: Could you please confirm whether my understanding is correct? In this work, the focus is not on providing a way to achieve data-faithful feature attribution explanations for the sake of explaining model outputs, but on *understanding* the relationship between data and target values. The model trained on instrumental variables explains this relationship as feature attribution applied to this model are data-faithful. Can it be seen as an analysis tool for phenomena, e.g. a disease as in the example you mentioned?
> >
> > Q6: Just a comment: Feature attribution in image classification usually treats each pixel as a feature, but I like your connection to lighting conditions. Perhaps data-faithful concept attribution fits the approach of confounders as well, what do you think?

---

> ### Author Response · Authors · 2024-08-09
> **Response to the official comment of Reviewer 6T6P**
>
> Thank you so much for your insightful follow-up questions and comments.
>
> * Q1: Yes, your understanding is correct.
>
> * Q2: Yes, your understanding is correct. Our proposed approach can be used as a tool for analyzing phenomena, making it particularly suitable in medical and financial fields.
>
> * Q6: Yes, data-faithful concept attribution is a suitable scenario for applying the approach of confounders, and we appreciate this idea. The example mentioned in the rebuttal can be effectively modeled by treating the concept of lighting conditions as an unobservable confounder.
>
>     Besides, we also think that data-faithful feature attribution is meaningful in the presence of pixel-level confounders. In adversarial attacks, a well-known example is adding invisible pixel-level noise to a panda image, leading to the model misclassifying the image as a gibbon [1]. When we conduct feature attribution on the image, those noisy pixels appear important, though humans cannot see their significance. If we could train a model to eliminate the influence of such noise, the attribution would better match the judgment of a human on the panda image and the shape of the panda. We think that there might be similar noise confounders existing in image classification. If we remove these confounders, the feature attribution results of the model would be more data-faithful, which might imply better model robustness. Of course, the example of data-faithful attribution in the image we mentioned above has some differences from our paper's focus. Firstly, the method for removing pixel-level confounders in images might not use instrumental variables. Secondly, data-faithful feature attribution, as we mentioned here, aims to help humans judge the reliability of a model, as in image classification, where model performance is usually the main concern. The performance-oriented goal is different from the goal of our paper on understanding the relationship between input features and target feature values. We believe this is an interesting topic that deserves more detailed thought in terms of motivation and methodology.
>
> We hope our responses have clarified your questions and helped improve your opinion of our work.
>
> Reference
>
> [1] Ian J. Goodfellow, Jonathon Shlens, and Christian Szegedy. Explaining and Harnessing Adversarial Examples. ICLR 2015.

---

> > ### Comment · Reviewer_6T6P · 2024-08-09
> >
> > Yes, thanks for the confirmation! I now understand the paper better and see the proposed method as a promising direction to recover causal effects in data. I'll raise my score from 4->7.
> >
> > I would like to ask the authors to please make the scope and problem clearer in the final version of the paper, in order to avoid confusion with explanation methods that aim at explaining model outputs (like my case).

---

> > > ### Author Response · Authors · 2024-08-09
> > >
> > > Thank you very much for your comments. We appreciate your insights and will certainly follow your suggestions to make the scope and problem clearer in the final version.

---

### Official Review · Reviewer_Ye8R · 2024-07-12

**Soundness:** 3
**Presentation:** 3
**Contribution:** 2
**Rating:** 5
**Confidence:** 2

**Summary:**

The paper addresses the challenge of unobservable confounders in feature attribution methods, which can lead to misinterpretations. The authors propose a novel approach called "data-faithful feature attribution," which trains models free of confounders using instrumental variables (IVs) to ensure that feature attributions are faithful to the data generation process.

**Strengths:**

1. The introduction of a confounder-free model training approach using instrumental variables is a significant advancement. This method addresses a critical gap in feature attribution by ensuring data fidelity.
2. The paper provides a comprehensive theoretical analysis that clearly explains the impact of unobservable confounders on feature attribution and how the proposed method mitigates this issue.
3. The authors validate their approach on both synthetic and real-world datasets, showing up to a 67% improvement in attribution accuracy over baseline methods. This demonstrates the practical applicability and effectiveness of the method.
4. The proposed method significantly reduces errors in feature attribution, ensuring that the contributions of input features to the target feature are accurately represented.

**Weaknesses:**

1. The two-stage training process and the use of advanced techniques like IVs may be complex to implement and require significant computational resources, potentially limiting the method's practical usability.
2. The theoretical derivations are based on the assumption that the influence of unobserved confounders on the target features is linear. This assumption may not hold in all real-world scenarios, limiting the method's applicability.

**Questions:**

1. How can one reliably identify suitable instrumental variables in practice? What criteria should be used to ensure their effectiveness?
2. How does the method perform with non-linear effects of unobserved confounders? Have you conducted any experiments or simulations to demonstrate its robustness in such scenarios?
3. Have you conducted sensitivity analyses to evaluate the robustness of the method under different conditions, such as varying degrees of distributional shifts or the quality of IVs?
4. How does the proposed method scale with larger datasets and higher-dimensional data? Are there any benchmarks or performance metrics available to demonstrate its scalability?

**Limitations:**

Please refer to the above questions.

---

> ### Author Rebuttal · Authors · 2024-08-06
>
> Thank you for your encouraging and insightful comments. Please find our response below.
>
> **Response to Weaknesses**
> * W1: The implementation and computation complexity of the two-stage training process is similar to the model training in the model-faithful feature attribution. In the first stage, we need to train a DNN classifier
> $\hat{M_\phi}$ if $\tilde{\boldsymbol{x}}$ is discrete (or mixture density network $\hat{M_\phi}$ if $\tilde{\boldsymbol{x}}$ is continuous) to approximate the distribution of unconfounded $\tilde{\boldsymbol{x}}$. Thus, the implementation is easy and computation complexity does not require significant computation resources in the first stage.  The second stage of model training is the same as the training of the predictive model in model-faithful feature attribution, except that the feature values of $\tilde{\boldsymbol{x}}$ are replaced with the re-estimated unconfounded values. The re-estimated unconfounded values are sampled from the first-stage trained model, the sampling has little influence on the implementation and computation complexity of the second-stage model training. Therefore, the two-stage training process does not limit the method's practical usability. We will clarify it in the paper.
>
> * W2: We conduct the theoretical derivations based on the assumption that the influence of unobserved confounders on the target features is linear. We can give an accurate formula for the error in the linear assumption while other types of influence are difficult to decouple in theoretical analysis. However, our method can still work well in the practical scenario when the effect of the confounders is not linear. In the real datasets Griliches76 dataset and the Angrist and Krueger dataset we used in Section 5.2, the confounder of Ability has a non-linear impact on income.  The experimental results still demonstrate significant effectiveness.
>
> **Response to Questions**
>
> * Q1: To identify suitable instrumental variables (IVs), we can first conduct a preliminary screening, such as LASSO regression, to assess the correlation between potential IVs and confounded features. Variables with non-zero coefficients may be correlated with $\tilde{\boldsymbol{x}}$ and are considered potential IVs. Next, we can use causal interventions to empirically verify these IVs, ensuring their effectiveness through statistical tests, such as checking if the F-statistic exceeds a threshold or using Hansen tests for validity. For a more in-depth study on IV identification, we recommend the seminal works [1][2][3]. Although identifying IVs is beyond this paper's scope, we will add a discussion to make this paper more self-contained and accessible to readers.
>
> * Q2: Our method can effectively address the non-linear effects of unobserved confounders. In experiments with real datasets, such as Griliches76 and Angrist and Krueger, the confounder Ability has a non-linear impact on income. For example, while factors like IQ positively correlate with income, the relationship is not linear. Although we assume linear effects of confounders for the convenience of theoretical analysis, our method remains effective for non-linear effects. This is because we re-estimate $\tilde{\boldsymbol{x}}$ using instrumental variables in the first stage, allowing us to obtain unconfounded $\tilde{\boldsymbol{x}}$ for attribution, regardless of how unobserved confounders influence $Y$.
>
> * Q3: In Section 5.1, we examined distribution shifts using synthetic datasets because real datasets do not allow control over external confounders and instrumental variables (IVs). In our synthetic datasets, we use $\rho$ to control the magnitude of unobserved confounders, $\tilde{\boldsymbol{x}}$ and the unobserved confounders are correlated. Consequently, the distribution of $\tilde{\boldsymbol{x}}$ also changes. Figures 1 and 2 demonstrate that our method's effectiveness increases with confounder influence. Notably, even at $\rho=0.2$, our method is effective.  Additionally, experiments show that higher-quality IVs reduce confounder-induced errors more effectively. For example, we test the average error of IV-SHAP on Dataset A when $\rho=0.2$ with $\psi \sim (0,0.5), \psi \sim (0,1), \psi \sim (0,2)$ where $\psi$ controls the correlation of IVs with $\tilde{\boldsymbol{x}}$. The results are shown in the table below.
>
>      |  Feature Deviation   | 0.125   | 0.250   | 0.375   | 0.500   |
>      |----------|---------|---------|---------|---------|
>      | $\psi \sim (0,0.5)$ | 0.034 | 0.174 | 0.268 | 0.397  |
>      | $\psi \sim (0,1)$ | 0.018 |  0.057 | 0.166 | 0.280  |
>      | $\psi \sim (0,2)$ | 0.012 | 0.039 | 0.0127 | 0.204  |
>
>
> * Q4: Our method scales well with larger datasets and higher-dimensional data. While larger datasets increase the training cost of the two-stage model, they do not complicate implementation. The training method costs twice as much as a typical predictive model, but remains manageable. Besides, the larger datasets increase the robustness and stability of the two-stage model. For higher-dimensional data, the computational demand increases, so we introduced approximation methods for IV-SHAP and IV-IG (see Appendix Section F), which we validated with the Spambase dataset. Extending our experiments to a 100-dimensional synthetic dataset, we found our method remained effective. Since data-faithful feature attribution is used mainly for tabular data to help understand feature relationships, and typically does not involve large dimensions. The average attribution ratio can still be used to evaluate scalability in the experimental section.
>
> References
>
> [1]Angrist, J. D., \& Imbens, G. W. Identification and estimation of local average treatment effects. Econometrica, 1994.
>
> [2]Baiocchi, M., Cheng, J., \& Small, D. S. Instrumental variable methods for causal inference. Statistics in Medicine, 2014.
>
> [3]Murray, M. P.  Avoiding invalid instruments and coping with weak instruments. Journal of Economic Perspective, 2006.

---

### Official Review · Reviewer_p3Si · 2024-07-13

**Soundness:** 3
**Presentation:** 3
**Contribution:** 3
**Rating:** 6
**Confidence:** 4

**Summary:**

This paper addresses the problem of estimating causal effects with feature attribution, by applying SHAP and Integrated Gradients (IG) to two-stage models with instrumental variables. On synthetic datasets, the proposed methods IV-SHAP and IV-IG can better recover the ground-truth causal effects, compared to SHAP and IG applied to predictive models. On real-world datasets, IV-SHAP and IV-IG are more aligned with prior knowledge than SHAP and IG.

**Strengths:**

- This paper addresses an important problem in the field of feature attribution: gaining causal insights from the attribution scores. Although it seems intuitive that a predictive model cannot provide causal insights when paired with feature attribution methods, this paper formally illustrates the need to account for confounding.
- The writing is clear. The pronounced focus on data fidelity in the introduction makes it clear that model-centric interpretation is not the goal of this paper. The conceptual examples are illuminating.

**Weaknesses:**

- It is unclear how the expectations with respect to $\hat{M}_{\phi}(\tilde{x} | \bar{x}_t, \psi_t)$ in Equations (4) and (6) are empirically computed in the experiments. The (approximate) computation can potentially be expensive.
- What is the impact on IV-SHAP and IV-IG when the three assumptions of instrumental variables are violated? The paper lacks a discussion on this.
- What is the impact on IV-SHAP and IV-IG when the input features are correlated? The paper lacks a discussion on this.

**Questions:**

- Besides SHAP and IG, is it appropriate to apply other feature attribution methods to a two-stage model in order to estimate causal effects?
- For the real-world datasets, what are the baseline values for IV-IG, and how are features removed in IV-SHAP?

Minor comments:
- There are typos in lines 172 and 178: $\frac{1}{\mathcal{N}}$ should be replaced with $\frac{1}{|\mathcal{N}|}$.
- I recommend omitting the $D$ in $\mathbb{E}_D$ of Proposition 2 for consistency.
- The use of Hoeffding's inequality in the proof for Lemma 7 require some boundedness condition on $h(u_i)$, which is reasonable to assume but should be mentioned nonetheless.

**Limitations:**

The limitations of the proposed approach itself seem adequately addressed.

---

> ### Author Rebuttal · Authors · 2024-08-06
>
> Thank you for your encouraging and insightful comments. Please find our response below.
>
> **Response to Weaknesses**
>
> * W1: For the case of discrete $\tilde{\boldsymbol{x}}$, $\hat{M_\phi}$ is trained as a DNN classifier with softmax output, where the output element represents the probability of the corresponding category. We can enumerate the probability of each category and compute the weighted sum of all categories when we need to compute the expectation term condition on  $\hat{M_\phi}\left(\tilde{\boldsymbol{x}} \mid \overline{\boldsymbol{x_t}}, \psi_t \right)$. The computation complexity is $O(c*n)$, where $c$ is the number of categories and $n$ is the number of sampled data tuples in model training. For the case of continuous $\tilde{\boldsymbol{x}}$, we implement a mixture density network to fit the distribution of $\hat{M_\phi}\left(\tilde{\boldsymbol{x}} \mid \overline{\boldsymbol{x_t}}, \psi_t\right)$. We use 20 random samples from the distribution output by the mixture density network to approximate the expectation, which has demonstrated great stability in our experiments. Thus, the computation complexity is $O(n)$. Both cases can be effectively computed given typical computational resources. We will add the details in the paper.
>
> * W2: The effectiveness of IV-SHAP and IV-IG may be reduced when the three assumptions of instrumental variables are violated. However, the extent of this reduction depends on how severely the assumptions are violated. To mitigate the influence of unobservable confounders, the instrumental variables need to have a more direct influence on $\tilde{X}$ while having less correlation with other features. Additionally, the instrumental variables should have little or no direct influence on $Y$. The instrumental variables are still effective at reducing the impact of unobservable confounders if the assumptions are only mildly violated. For example, in our experiment with the Griliches76 dataset, the chosen instrumental variable, Parental Education, has a weak correlation with the unobservable confounder, Ability, as we analyzed in Appendix G.2. However, the influence of parental education on child education is much stronger, as demonstrated by the statistical characteristics of the real dataset in Appendix G.2. Despite this mild violation, the effectiveness of IV-SHAP and IV-IG remains significant. We will add a discussion in the paper/
>
> * W3: The correlation of input features may affect the data-faithfulness of IV-SHAP and IV-IG. We can combine methods which deal with the correlated input features to the two-stage model to better capture these correlations. For example, on-manifold Shapley [1] can be used to account for feature correlations, while causal Shapley [2] can be applied if the causal structure of the input features is known. We note that our primary focus in this paper is on addressing the influence of unobservable confounders. Therefore, we chose the two most representative feature attribution methods, SHAP and IG, to formulate the problem. We will include this discussion in the paper.
>
> **Response to Questions**
>
> * Q1: Other feature attribution methods can be applied according to the demands. For example, when considering the correlation between input features, we can apply on-manifold Shapley or causal Shapley to our two-stage model if sufficient prior knowledge is available. Our methods focus on addressing unobservable confounders by re-estimating $\tilde{\boldsymbol{x}}$. Therefore, we can combine other feature attribution methods to handle the input features after re-estimating $\tilde{\boldsymbol{x}}$ with the two-stage model.
>
> * Q2: We experimented with multiple baseline values by subtracting specific values from the features $\tilde{\boldsymbol{x}}$ and $\overline{\boldsymbol{x}}$ in both synthetic and real datasets. Lines 283-286 and 311-312 in Section 5 provide the details of experimental setting. The removed features are replaced with the average values from the dataset in IV-SHAP, which is consistent with the approach in SHAP.
>
> **Response to Minor comments**
>
> * M1: Fixed. Thanks!
>
> * M2: We have omitted the $D$ following your suggestion.
>
> * M3: $h(u_i)$ is the weighted gradients of a deep neural network, which is continuously differentiable and bounded in our setting. We will clarify it in the paper.
>
>
> References
>
> [1]Frye C, de Mijolla D, Begley T, et al. Shapley explainability on the data manifold[C]. International Conference on Learning Representations (ICLR).
>
> [2]Heskes T, Sijben E, Bucur I G, et al. Causal Shapley values: Exploiting causal knowledge to explain individual predictions of complex models[J]. Advances in neural information processing systems (NeurIPS).

---

> > ### Comment · Reviewer_p3Si · 2024-08-13
> >
> > Thank you for clarifying my questions. I appreciate the authors for explaining that the performance of IV-SHAP and IV-IG can depend on (i) whether the three assumptions of instrumental variables are violated and (ii) the correlations between input features. I don't consider these as weaknesses specific to IV-SHAP and IV-IG, but rather general problems in causal inference. I encourage the authors to highlight the importance of (i) and (ii) so readers have such awareness when using IV-SHAP and IV-IG. For example, ablation studies with synthetic datasets can help readers assess the impact of (i) and (ii) on the effectiveness of IV-SHAP and IV-IG. Overall, the authors have addressed my questions, so I maintain my original score of 6.

---

> ### Author Response · Authors · 2024-08-13
>
> Thank you very much for your valuable comments. We appreciate your emphasis on the assumptions of instrumental variables and correlations of input features, which we agree are crucial factors for the effectiveness of IV-SHAP and IV-IG. We will make sure to highlight these points in the final version.

---

### Author Rebuttal · Authors · 2024-08-06

We are very grateful to the reviewers for their encouraging and insightful comments. To address the concerns, we provide detailed point-to-point responses as follows.

---

### Decision · Program_Chairs · 2024-09-25

**Decision:**

Accept (poster)

**Comment:**

The paper makes an important contribution in understanding the causal attribution that drives an outcome variable. Rather than standard attribution scores that explain a predictive model, the authors propose an IV-based method that explains the true outcome variable. The method is motivated through a theoretical analysis of the effect of unobserved confounder on attribution scores. Reviewers agreed that the technical contribution and insight is note-worthy and therefore I'm recommending Accept. However, they also have two suggestions for improving the clarity of the paper:
1) It will be helpful to make it clear in the abstract and introduction that the goal is to explain the true outcome variable (not a given ML model)
2) The limitations in Appendix A and the limitations discussed in the rebuttal period should be included in the main text.